# Pharmacological inhibition of adipose triglyceride lipase corrects high-fat diet-induced insulin resistance and hepatosteatosis in mice

Martina Schweiger[1],*, Matthias Romauch[1],*, Renate Schreiber[1], Gernot F. Grabner[1], Sabrina Hütter[1], Petra Kotzbeck[1], Pia Benedikt[1], Thomas O. Eichmann[1], Sohsuke Yamada[1], Oskar Knittelfelder[1], Clemens Diwoky[1], Carina Doler[2], Nicole Mayer[2], Werner De Cecco[3], Rolf Breinbauer[2], Robert Zimmermann[1] & Rudolf Zechner[1]

Elevated circulating fatty acids (FAs) contribute to the development of obesity-associated metabolic complications such as insulin resistance (IR) and non-alcoholic fatty liver disease (NAFLD). Hence, reducing adipose tissue lipolysis to diminish the mobilization of FAs and lower their respective plasma concentrations represents a potential treatment strategy to counteract obesity-associated disorders. Here we show that specific inhibition of adipose triglyceride lipase (Atgl) with the chemical inhibitor, Atglistatin, effectively reduces adipose tissue lipolysis, weight gain, IR and NAFLD in mice fed a high-fat diet. Importantly, even long-term treatment does not lead to lipid accumulation in ectopic tissues such as the skeletal muscle or heart. Thus, the severe cardiac steatosis and cardiomyopathy that is observed in genetic models of *Atgl* deficiency does not occur in Atglistatin-treated mice. Our data validate the pharmacological inhibition of Atgl as a potentially powerful therapeutic strategy to treat obesity and associated metabolic disorders.

[1] Institute of Molecular Biosciences, University of Graz, Heinrichstrasse 31, 8010 Graz, Austria. [2] Institute of Organic Chemistry, Graz University of Technology, Stremayrgasse 9, 8010 Graz, Austria. [3] Institute of Chemistry, University of Graz, Heinrichstrasse 28, 8010 Graz, Austria. * These authors contributed equally to this work. Correspondence and requests for materials should be addressed to M.S. (email: tina.schweiger@uni-graz.at) or to R. Zimmermann (email: robert.zimmermannuni-graz.at) or to R. Zechner (email: rudolf.zechner@uni-graz.at).

Obesity and its metabolic consequences including insulin resistance (IR) and non-alcoholic fatty liver disease (NAFLD) are global health threats with increasing prevalence[1]. Effective prevention and treatment strategies are needed to halt this detrimental development including the option of pharmacological interventions. To date, however, few potent and safe therapeutics that promote weight loss and improve metabolic health are available[2].

From a reductionist's point of view, obesity in mammals results from an imbalance between the rates of fat synthesis and fat catabolism in white adipose tissue (WAT). This concept finds strong support in studies with stable isotopes showing that expansion of fat mass in obese individuals results from increased triglyceride (TG) synthesis and decreased TG breakdown in WAT[3]. TG breakdown is defined as the enzymatic cleavage of TGs and the formation of fatty acids (FAs) and glycerol. This process called lipolysis requires at least three distinct hydrolases—adipose TG lipase (Atgl, officially annotated as Pnpla2), hormone-sensitive lipase (Hsl) and monoglyceride lipase, which consecutively release three FAs from the glycerol backbone[4]. Unexpectedly, humans and mice lacking ATGL, HSL or monoglyceride lipase are not or only moderately obese and it is therefore unclear whether WAT lipolysis is a major 'driver' for obesity[5–7].

In addition, although fat mass and in particular intra-abdominal, 'visceral' fat strongly correlates with both IR and NAFLD, the causative basis for this connection and the role of lipolysis in it remains a matter of extensive debate. A popular hypothesis proposes that increased WAT lipolysis generates excessive amounts of circulating FAs, which are subsequently absorbed by the liver and other ectopic tissues, and cause NAFLD and IR[8]. In this scenario, NAFLD and IR result from FA-induced lipotoxicity where FAs undergo transformation into TGs and bioactive lipid species (for example, diacylglycerols, ceramides or prostaglandins). Recent data in humans harbouring mutations in the gene encoding perilipin-1 strongly support this concept. Gandotra et al.[9] demonstrated that WAT lipolysis is hyperactive in these patients due to a constitutive activation of ATGL. Unrestrained lipolysis leads to the development of severe IR, type-2 diabetes and NAFLD in these patients[9,10].

The lipotoxicity model suggests that inhibition of lipolysis could be an attractive approach to lower plasma FA concentrations, thereby reducing the availability of FAs and their lipotoxic impact in ectopic tissues. Nicotinic acid (niacin) has been used in humans to lower plasma lipids by targeting lipolysis via a G-protein-coupled receptor. However, the drug has several off-target side effects limiting its long-term applicability and acceptance by patients[11]. Furthermore, small-molecule inhibitors for Hsl have been tested in mice. Hsl inhibition resulted in improved insulin sensitivity but did not affect body weight, fat mass and WAT inflammation in mice fed a high caloric diet[12]. With regard to Atgl, the characterization of mouse models lacking Atgl in specific or all tissues of the body rather discouraged than encouraged the development of chemical inhibitors for the enzyme[5,13]. Although, global Atgl-deficient mice exhibited improved insulin sensitivity and glucose tolerance, they suffered from systemic TG accumulation and lethal cardiomyopathy characterized by severe cardiac steatosis, mitochondrial and respiratory dysfunction, and organ failure[5,14]. Similarly, human patients with loss-of-function mutations in the ATGL gene develop neutral lipid storage disease with myopathy defined by an accumulation of fat in multiple tissues and the occurrence of severe skeletal and cardiomyopathy[13]. More recent studies in rescued mice expressing the enzyme exclusively in the heart[14,15] were more encouraging by showing that Atgl deficiency improves glucose tolerance and insulin sensitivity on chow and high-fat diet (HFD) and protects the animals from HFD-induced obesity[5,16]. Similar improvements in insulin sensitivity were observed in mice lacking Atgl exclusively in the adipose tissue[17], suggesting that with the exception in cardiac physiology, inhibition of Atgl may generate a beneficial metabolic phenotype.

The current study addressed the question whether inhibition of Atgl by the competitive, small molecule inhibitor Atglistatin[18] can prevent or cure HFD-induced metabolic derangements. We show that inhibitor treatment effectively improves HFD-induced obesity, IR and liver steatosis in mice. The drug predominantly targets adipose tissue and the liver, and therefore does not cause cardiac lipid accumulation or cardiomyopathy even after long-term treatment. Thus, if—similar to Atglistatin—medication does not target cardiac lipolysis, the chronic inhibition of Atgl may represent an attractive means to treat metabolic disorders.

## Results

**Atglistatin transiently inhibits murine but not human lipolysis.** We recently reported that a single application of Atglistatin in mice transiently inhibits lipolysis for ~8 h when applied intraperitoneally (i.p.) or by gavage[18]. To assess whether Atglistatin reduces plasma FA concentrations when the inhibitor is administered via food uptake, C57Bl6 mice were fed a HFD (45 kJ% fat; 22.1 kJ g$^{-1}$) for 50 days. Before drawing the blood sample, mice were fasted for 7 h and then re-fed a HFD with or without Atglistatin (2 mmol kg$^{-1}$ diet) for 2 h. During this re-feeding period, Atglistatin-treated (ATGLi) mice consumed 0.8 ± 0.2 g HFD equivalent to 1.6 μmol Atglistatin representing an effective inhibitory concentration (56 μmol kg$^{-1}$)[18]. Control mice ate the same amount of food without the inhibitor. Two hours after the start of the re-feeding period, ATGLi animals exhibited 47% lower plasma FA levels compared with control animals (Fig. 1a). This decrease disappeared after a second, subsequent fasting period of 8 h, suggesting that the effect of Atglistatin on the lipolytic FA release in WAT was transient. To analyse WAT lipolysis, secretion rates of FAs from gonadal (g)WAT explants were determined ex vivo. Two hours after the start of the re-feeding period, gWAT explants from ATGLi animals released 57% less FAs into the medium than gWAT explants from control animals (Fig. 1b). Addition of Atglistatin to these gWAT explants reduced FA release from control samples by 51% but had no effect on FA release from samples of ATGLi animals (Fig. 1b). When mice were again fasted for 8 h after the 2 h re-feeding period, FA release from gWAT explants was similar in ATGLi and control mice, despite increased Atgl and slightly reduced Hsl protein levels in ATGLi WAT (Fig. 1c and Supplementary Fig. 1). Addition of Atglistatin to gWAT explants reduced FA release to the same extent in the explants of both groups (−45%) (Fig. 1c). Thus, Atglistatin causes a transient reduction in lipolysis as long as the animals have access to food and does not inhibit WAT lipolysis after food deprivation for 8 h or longer.

To clarify whether Atglistatin is also able to inhibit human ATGL, we performed in vitro TG hydrolase activity assays using recombinant human and murine CGI-58 activated ATGL in the presence and absence of Atglistatin (Fig. 1d). Although 200 μM Atglistatin efficiently inhibited mouse Atgl activity (−95%), human ATGL was barely affected (−10%). Consistent with this result, Atglistatin effectively inhibited the release of FA from murine 3T3-L1 adipocytes (−98%) at a concentration of 50 μM, but failed to inhibit FA release from human Simpson-Golabi-Behmel-Syndrome (SGBS) adipocytes (Fig. 1e).

**Pharmacological inhibition of Atgl ameliorates HFD-induced obesity.** To investigate the metabolic effects of chronic Atglistatin

feeding during high caloric intake, 6 weeks old male C57Bl6 mice were fed a HFD for 50 days followed by another 50 days HFD feeding in the presence or in the absence of Atglistatin. Control animals gained weight ($+5.3$ g) due to an increase of fat mass ($+5.1$ g), whereas ATGLi animals lost 1.4 g body weight (Fig. 1f,g). This reduction in weight was entirely due to a decrease in adipose mass ($-1.3$ g), whereas lean mass remained unchanged (Fig. 1g). Organ weights of inguinal (i)WAT, gWAT and interscapular brown adipose tissue (iBAT) were reduced by

70%, 76% and 42%, respectively, in ATGLi animals compared with controls (Fig. 1h). The resistance to HFD-induced obesity in ATGLi mice was accompanied by a 90% decrease and 16% increase in plasma concentrations of leptin and adiponectin, respectively (Table 1). Furthermore, Atglistatin significantly reduced fasting (5 h) plasma levels of FA ($-13$%), glycerol ($-33$%), TG ($-16$%) and cholesterol ($-25$%), whereas no differences were detected for plasma levels of lactate and β-hydroxybutyrate (Table 1).

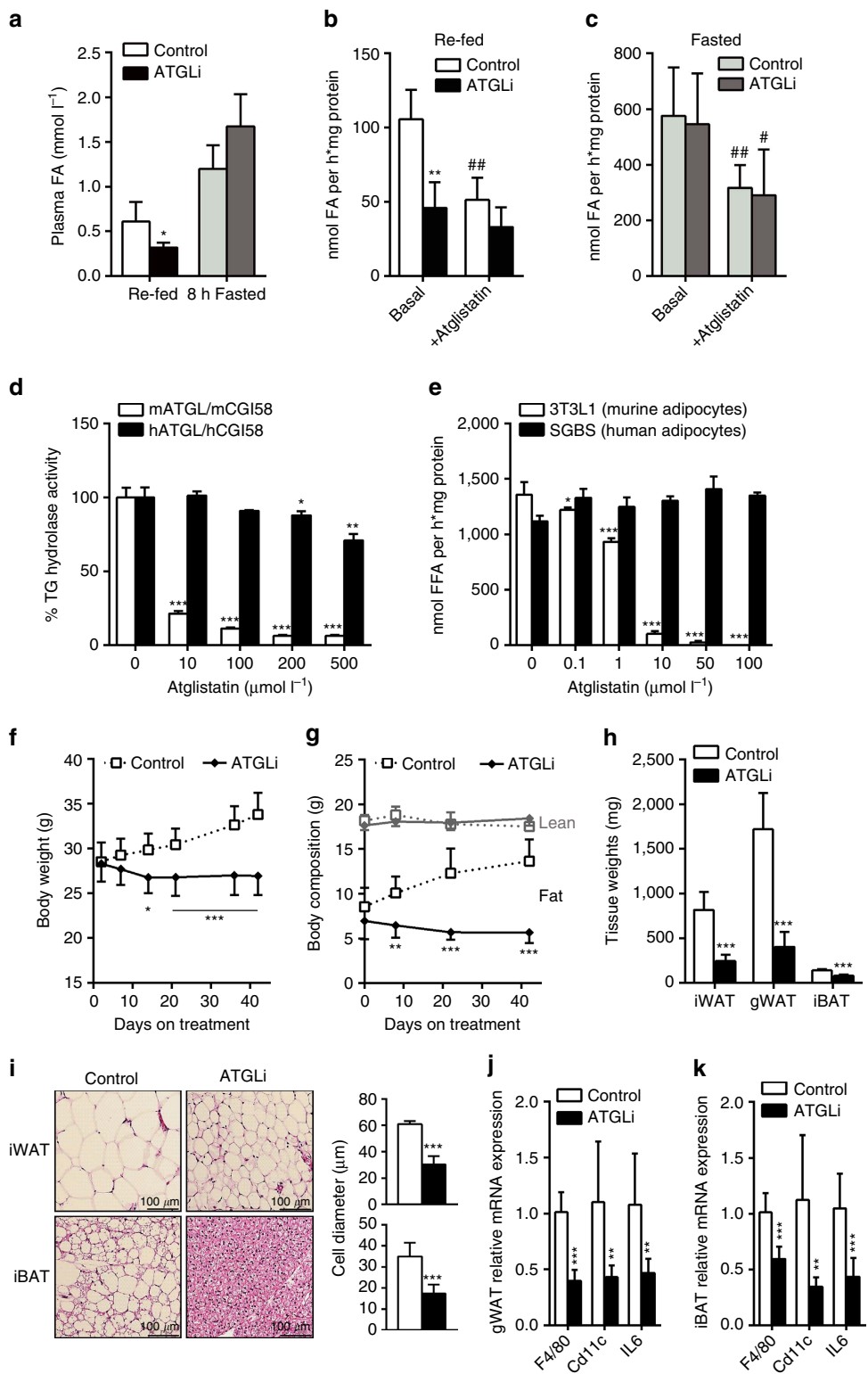

Adipocyte diameters were markedly reduced in iWAT (−50%) and iBAT (−51%) of ATGLi versus non-treated mice (Fig. 1i). As adipocyte hypertrophy is commonly associated with the infiltration of immune cells, we assessed whether the lean phenotype of ATGLi mice protects the animals from this inflammatory response. Reverse transcriptase–quantitative PCR (qPCR) analysis of RNA samples from gWAT and iBAT revealed a 60% lower expression of the macrophage-specific markers F4/80 and Cd11c, and a 50% reduced expression of the pro-inflammatory cytokine interleukin (Il)-6 (Fig. 1j,k). Reduced Il-6 expression in adipose tissues was associated with a 45% decrease in Il-6 plasma concentrations in ATGLi mice compared with control mice (Table 1), arguing for decreased inflammation in mice in response to inhibitor treatment.

To delineate whether Atglistatin exerts its anti-obesity effects by an Atgl-dependent or -independent 'off-target' effect, we fed 'cardiac-rescued' Atgl-deficient (AKO/cTg) and control animals a HFD ± Atglistatin. AKO/cTg mice express an α-myosin heavy chain-driven Atgl transgene on an Atgl-deficient background. Accordingly, these mice express Atgl exclusively in cardiac myocytes, but lack the enzyme in all other tissues of the body[14,15]. This genetic manipulation prevents the lethal cardiomyopathy observed in global Atgl deficiency. HFD-fed wild-type (WT) mice expressing the Atgl transgene on a WT background served as control (WT/cTg). After 35 days, ATGLi WT/cTg mice had 8.7 g lower body weight than untreated WT/cTg mice and exhibited drastically reduced adipose tissue depots (iWAT −60%, gWAT −80% and iBAT −55%) (Supplementary Fig. 2a,b). ATGLi AKO/cTg mice exhibited 2.4 g less body weight than non-treated AKO/cTg animals, and iWAT and iBAT remained unaffected, whereas gWAT exhibited a 28% reduced size compared with non-treated AKO/cTg mice. Thus, the resistance to an HFD-induced increase in iWAT entirely depends on the inhibition of Atgl. Whether the decrease in gWAT in ATGLi AKO/cTg mice and hence the slight reduction in body weight relates to a minor off-target effect remains to be determined.

**Atglistatin causes hypophagia and reduced lipid deposition in WAT.** To evaluate the potential mechanisms causing obesity resistance in ATGLi mice, we analysed parameters of energy balance. When calculated per mouse, energy expenditure (EE) was reduced in ATGLi compared with control animals. However, linear regression analysis revealed a strong linear correlation between EE and body weight (Fig. 2a; $R^2 = 0.9$), as well as EE and fat mass (Supplementary Fig. 3; $R^2 = 0.9$) in HFD-fed mice. As body weight and fat mass, but not lean mass, differ between control and ATGLi mice, we adjusted EE for body weight or fat mass using analysis of covariance (ANCOVA) (insert Fig. 2a and insert Supplementary Fig. 3). The results revealed that EE in ATGLi mice is reduced because of reduced fat mass but Atglistatin treatment *per se* does not affect EE. However, we noticed a

significant change in food uptake. During metabolic monitoring, ATGLi animals ate 10% less than control animals resulting from 32% reduced food intake during dark phase and a 16% increased food intake during light phase (Fig. 2c). These differences reflect a sharp increase in food intake of ATGLi animals already at 16:00 h (light phase), whereas control mice started to eat at 21:00 h (dark phase) (Fig. 2d). This time shift in food intake is also reflected by an increase in respiratory exchange ratio (RER) at earlier time points in ATGLi animals (17:00 h) compared with controls (22:00 h) (Fig. 2e). Moreover, means of total and light-phase RER were significantly elevated in ATGLi mice, indicating increased carbohydrate combustion due to inhibited Atgl-mediated lipolysis (Fig. 2f). However, no differences in RER where observed during times of low food intake (that is, low Atglistatin uptake) again indicating time-limited inhibition of lipolysis. Water intake and locomotor activity was not affected by Atglistatin treatment (Fig. 2b,g).

To determine to what extent reduced food intake accounts for the decreased WAT mass and body weight in ATGLi mice, we performed pair-feeding experiments for 68 days. The amount of food for control animals was adjusted to the food consumed by ATGLi animals. Food consumption in both groups of mice was lowest during the first 10 days (1.7 g per mouse*day) and then gradually increased to 2.5 g per mouse*day (Fig. 3a). During the first 10 days, both groups lost weight (−4.8 and −5.3 g). When food intake increased to 2.5 g per day pair-fed control animals started to gain weight (∼0.1 g per day), whereas ATGLi animals further reduced their body weight by −0.07 g per day (Fig. 3b). Thus, reduced food intake contributes but does not fully account for the decreased adiposity in ATGLi mice.

The drastic morphological difference of BAT (Fig. 1i) between ATGLi and control mice prompted us to investigate whether increased thermogenesis contributes to reduced weight gain. Ucp1, CideA and Pgc1α messenger RNA levels in iBAT were similar in control and ATGLi mice (Supplementary Fig. 4a). In contrast, iBAT of ATGLi mice had higher Ucp1 protein levels than untreated mice. However, this did not translate into increased core body temperature at ambient housing temperatures (Supplementary Fig. 4b,c). At thermoneutral housing temperatures (30 °C) where BAT activity is low, weight loss of ATGLi mice was higher (−17%) than in pair-fed control mice (−6.5%) (Supplementary Fig. 4d). Similar as in chow-fed animals, this decrease in body weight was again entirely due to a reduction in fat mass with no changes in lean mass of ATGLi mice (Supplementary Fig. 4e,f,g). iBAT of ATGLi mice in thermoneutrality consisted of smaller adipocytes containing smaller lipid droplets than in control mice, similar as observed at ambient temperatures (Supplementary Fig. 4h). Taken together, these data indicate that the loss of WAT mass upon Atglistatin treatment is not caused by an activation of BAT activity or increased thermogenesis.

**Figure 1 | Atglistatin transiently inhibits lipolysis and protects from HFD-induced obesity.** Six weeks old male C57Bl6J mice were fed a HFD (45 kJ% fat; 22.1 kJ g$^{-1}$) for 50 days. Thereafter, mice were fasted for 7 h and then re-fed a HFD with or without Atglistatin (2 mmol kg$^{-1}$ diet) for 2 h followed by a second, subsequent fasting period of 8 h. (**a**) Plasma FA levels were determined in the 2 h re-fed and 8 h fasted state (n = 5). (**b**) FA release from gonadal adipose tissue explants of re-fed and (**c**) 8 h-fasted mice (n = 5 per group). (**d,e**) Atglistatin does not inhibit human adipocyte lipolysis. (**d**) TG hydrolase activity was assessed in COS-7 lysates of cells overexpressing human and murine ATGL and CGI-58, respectively, in the presence and absence of the indicated concentrations of Atglistatin. (**e**) SGBS and 3T3-L1 preadipocytes were differentiated to adipocytes. Then, cells were preincubated with the indicated concentrations of Atglistatin for 2 h. Thereafter, the medium was replaced by DMEM containing 2% BSA, 10 µM Forskolin and the indicated concentrations of Atglistatin for 1 h. The release of FA in the medium was determined and calculated per mg cell protein. (**f–k**) Mice were fed a HFD for 50 days, followed by HFD-feeding in the presence or absence of Atglistatin for another 50 days. (**f**) Body weight, and (**g**) fat and lean mass development (n = 7 per group). Adipose tissue depots, inguinal (i)WAT, gonadal (g)WAT, and interscapular (i)BAT were analysed for their (**h**) weight and (**i**) adipocyte size. (**j,k**) mRNA expression of IL-6 and macrophage markers F4/80 and Cd11c was assessed in gWAT and iBAT of re-fed mice (n = 5 per group). Data represent mean ± s.d. Statistical significance was determined by Student's two-tailed t-test. For analysis of multiple measurements, we performed one-way analysis of variance (ANOVA) followed by Bonferroni *post-hoc* test; *P < 0.05, **P < 0.01 and ***P < 0.001 for control versus ATGLi; #P < 0.05 and ##P < 0.01 for basal versus + Atglistatin.

**Table 1 | Plasma parameters of mice fed a HFD in the presence and absence of Atglistatin.**

| | HFD 5h fasted | | HFD re-fed | |
|---|---|---|---|---|
| | Control | Atglistatin | Control | Atglistatin |
| Leptin (ng ml$^{-1}$) | 44.09 ± 18.8 | 1.56 ± 0.45*** | 40.82 ± 14.2 | 6.41 ± 2.08*** |
| IL-6 (pg ml$^{-1}$) | 0.8 ± 0.18 | 0.44 ± 0.25* | n.a. | n.a. |
| Adiponectin (µg ml$^{-1}$) | 24.6 ± 0.73 | 28.6 ± 3.24* | n.a. | n.a. |
| Insulin (ng ml$^{-1}$) | 1.55 ± 0.58 | 0.44 ± 0.17*** | 2.33 ± 0.97 | 1.21 ± 0.67* |
| Glucose (mg dl$^{-1}$) | 189 ± 10.8 | 169 ± 3.9* | 221 ± 13.1 | 191 ± 21.4** |
| QUICKI | 0.26 ± 0.01 | 0.31 ± 0.02*** | n.a. | n.a. |
| FA (mmol l$^{-1}$) | 0.69 ± 0.09 | 0.6 ± 0.03* | 0.48 ± 0.05 | 0.31 ± 0.09** |
| Glycerol (mmol l$^{-1}$) | 0.21 ± 0.02 | 0.14 ± 0.03* | 0.3 ± 0.1 | 0.21 ± 0.06* |
| TG (mmol l$^{-1}$) | 0.75 ± 0.12 | 0.63 ± 0.07* | 0.56 ± 0.19 | 1.36 ± 0.36* |
| Cholesterol (mg ml$^{-1}$) | 2.48 ± 0.05 | 1.85 ± 0.22** | 2.45 ± 0.29 | 1.76 ± 0.27*** |
| Lactate (mmol l$^{-1}$) | 3.0 ± 0.42 | 2.6 ± 0.99 | 3.0 ± 1.01 | 3.4 ± 0.62 |
| β-HB (mmol l$^{-1}$) | 0.16 ± 0.07 | 0.16 ± 0.05 | 0.1 ± 0.02 | 0.13 0.04 |

β-HB, β-hydroxybutyrate; FA, fatty acid; HFD, high-fat diet; IL-6, interleukin-6; TG, triacylglycerol
Six weeks old male C57Bl6J mice were fed a HFD (45 kJ% fat; 22.1 kJ g$^{-1}$) for 50 days. Thereafter, mice were fed a HFD in the presence and absence of Atglistatin for 50 days. Blood was collected by retro-orbital puncture from isoflurane-anaesthetized animals. Leptin, IL-6, adiponectin and insulin levels were determined using ELISA. FA, glycerol, TG, cholesterol, lactate and β-HB were measured using commercial kits. n = 6–10 per group. Data represent mean ± s.d. *P < 0.05, **P < 0.01 and ***P < 0.001; statistical significance was determined by two-tailed Student's t-test.

Next, we determined whether differences in intestinal food absorption contribute to the differences in HFD-induced obesity of ATGLi versus control mice. Atgl is prominently expressed in intestinal mucosa cells and may affect fat absorption and chylomicron formation and secretion[19]. Gross faeces output mass on three consecutive days was identical in ATGLi and control mice (Fig. 3c). However, calorimetric analysis revealed that faeces of ATGLi animals contained 2.8 kJ g$^{-1}$ more energy than faeces of control animals (Fig. 3d). To investigate whether this difference in faecal energy output is due to reduced intestinal lipid absorption, we determined intestinal absorption of dietary fat using the non-hydrolysable and non-absorbable sucrose polybehenate as internal standard in the HFD[20]. We found a slight but significantly reduced lipid absorption (−3.8%) in ATGLi animals compared with controls (Fig. 3e). This indicates that reduced lipid absorption also contributes to decreased obesity in ATGLi mice. Calculation of the average daily energy intake and output revealed that although ATGLi animals consume less food (−7 kJ per day) and lose more energy to the faeces (+1.3 kJ per day) compared with controls, these differences cannot fully explain the obesity-resistant phenotype of ATGLi mice (Supplementary Fig. 5).

Interestingly, despite slightly reduced lipid absorption, postprandial TG levels were 2.4-fold higher in ATGLi mice than in controls (Table 1). Oral lipid tolerance tests revealed that ATGLi animals exhibit 2.3-fold and 2.6-fold higher plasma TG concentrations than control animals 1 and 2 h after oil gavage, respectively (Fig. 3f). Delayed clearance of post-prandial fat concurred with a marked reduction of lipoprotein lipase (Lpl) mRNA levels (−43%) and enzyme activity (−87%) in gWAT of ATGLi mice after re-feeding (Fig. 3g,h). Decreased Lpl-mediated FA import into gWAT was accompanied by a decrease of Dgat2 mRNA (−64%) coding for the rate-limiting enzyme in TG synthesis in adipose tissue (Fig. 3h). Besides Lpl and Dgat2, both bona fide target genes of peroxisome proliferator-activated receptor gamma (PPARγ), also the expression Angptl4, Cd36 and G0s2 was reduced in WAT of ATGLi mice (Fig. 3h), suggesting that the ATGLi-mediated inhibition of lipolysis leads to a suppression of PPARγ target gene expression. These findings concur with our recent results[16] showing that genetic Atgl deficiency causes decreased TG accumulation in WAT due to suppressed PPARγ signalling and suggest that both genetic and pharmacological inhibition of

lipolysis lead to resistance to HFD-induced obesity via decreasing food intake and lowering TG synthesis and lipid deposition in WAT.

**Atglistatin-treatment improves glucose homeostasis.** To investigate how the chronic inhibition of Atgl-mediated lipolysis affects insulin sensitivity and glucose tolerance, mice received a HFD for 50 days followed by another 50 days HFD feeding in the presence or absence of Atglistatin. Moderately fasted (5 h) ATGLi mice displayed reduced plasma levels for glucose (−10%) and insulin (−71%), as well as an increased QUICK index (+20%) compared with control mice (Table 1). After re-feeding, plasma insulin concentrations increased in control and ATGLi animals but still remained 48% lower in ATGLi than in control mice, indicating intact but reduced insulin secretion from pancreatic β-cells. Despite decreased plasma insulin concentrations, i.p. insulin tolerance test (ITT, Fig. 4a) and glucose tolerance test (GTT, Fig. 4b) revealed that ATGLi animals cleared plasma glucose significantly more efficiently after insulin (−30%), as well as after glucose (−23%) injection (as estimated by the area under the clearance curve, Fig. 4a,b inserts) compared with HFD-fed control animals, indicating higher insulin and glucose tolerance. Comparing insulin sensitivity (Supplementary Fig. 6a) and glucose tolerance (Supplementary Fig. 6b) of HFD-fed mice before (0 weeks) and 6 weeks after Atglistatin treatment clearly showed an improvement of glucose clearance in response to drug administration.

**Atglistatin treatment reduces body weight and IR in ob/ob mice.** To additionally investigate whether inhibition of lipolysis by Atglistatin affects body weight and glucose homeostasis in a genetic mouse model of severe obesity and IR, we measured food intake, body weight and parameters of glucose homeostasis in ob/ob and WT mice fed a chow diet ± Atglistatin. During the course of 40 days of dietary intervention, WT mice gained 0.125 g body weight per day, whereas ATGLi animals only gained 0.025 g per day, despite the same amount of food intake (Supplementary Fig. 7a,b). Hence, also on chow diet Atglistatin reduces weight gain, although the effect is less pronounced than in animals on HFD. ob/ob mice gained 0.5 g per day body weight (Fig. 4c). This increase was attenuated by 30% in ATGLi ob/ob mice (+0.35 g per day), which can at least partially be explained by reduced food intake (Fig. 4c,d). Similar to

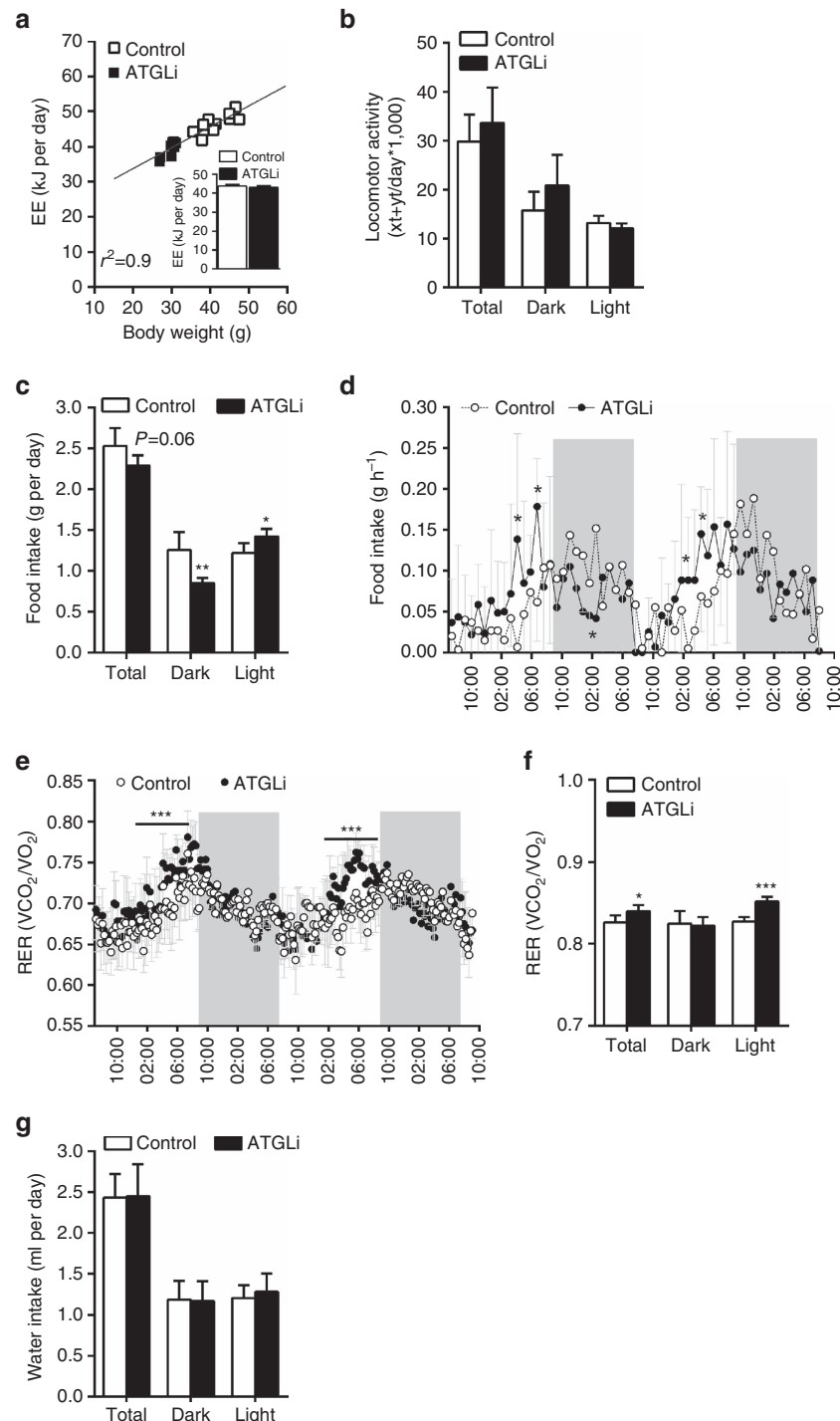

**Figure 2 | Metabolic characterization of mice fed a HFD in the presence and absence of Atglistatin.** Six weeks old male C57Bl6J mice were fed a HFD (45 kJ% fat; 22.1 kJ g$^{-1}$) for 50 days. Thereafter, mice were fed a HFD in the presence and absence of Atglistatin for 30 days. Metabolic phenotyping was performed using a laboratory animal monitoring system (LabMaster, TSE Systems, n = 6 per group). Mice were familiarized with the metabolic cages for 3 days prior measurement. (**a**) EE was calculated using the formula: EE(kJ per day) = 15,818*VO$_2$ + 5,176*VCO$_2$/1,000*24 and is plotted against body weight. Linear regression analysis was performed using GraphPad prism software. Data represent single mice. (**a**, insert) EE is expressed as adjusted means based on a normalized mouse weight of 35.73 g determined using ANCOVA, P = 0.94. (**b**) Locomotor activity per day. (**c**) Food intake per mouse and day, and (**d**) per mouse and hour during the course of the day. (**e**) Respiratory exchange ratio (CO$_2$/O$_2$) during the course of the day and (**f**) per day. Mice were kept on a 14 h light and 10 h dark cycle. Grey bars indicate dark phases. (**g**) Water intake per mouse and day. Data represent mean ± s.d. Statistical significance between control and ATGLi was determined by Student's two-tailed t-test; *P < 0.05, **P < 0.01 and ***P < 0.001.

HFD-fed WT mice, Atgl inhibition improved glucose clearance in *ob/ob* mice in GTT (area under the curve: −40%; Fig. 4e) and ITT (area under the curve: −40%; Fig. 4f) experiments. In contrast, glucose and insulin tolerance was unchanged in chow-fed ATGLi mice compared with control mice (Supplementary Fig. 7c,d). Together, these findings suggest that the inhibition of lipolysis improves glucose homeostasis in both dietary and genetic models of IR.

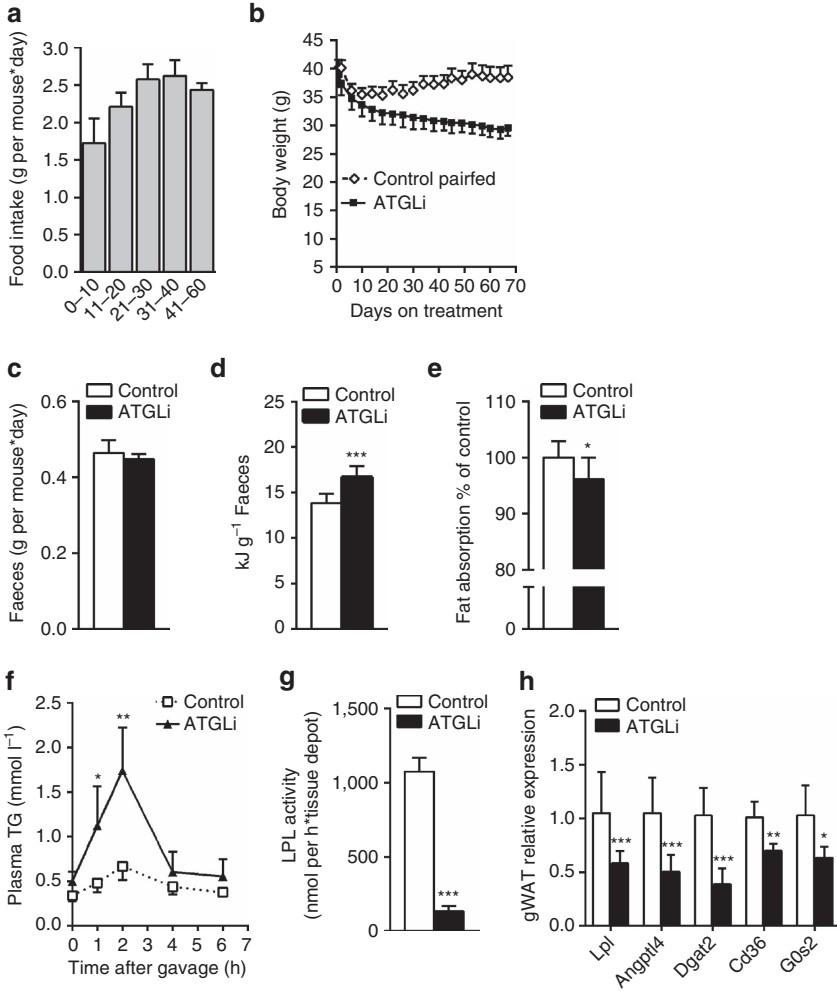

**Figure 3 | Hypophagia and reduced lipid deposition in WAT contribute to the obesity resistant phenotype.** (**a**) Six weeks old male C57Bl6J mice were fed a HFD (45 kJ% fat; 22.1 kJ g$^{-1}$) for 50 days. Thereafter, mice were fed a HFD in the presence or absence of Atglistatin for 68 days. Daily food intake in single housed mice was determined by subtracting food uneaten from food given to the bottom of the cages. Pair-feeding was performed by giving control mice the same amount of food ATGLi animals have eaten the day before ( ± 0.1 g; $n = 5$ per group). (**b**) Body weight of pair-fed and *ad libitum*-fed animals. (**c**) Faeces output was determined on 3 consecutive days ($n = 8$ per group). (**d**) Faeces of mice were sampled on 3 consecutive days and analysed for the excreted energy using a bomb calorimeter ($n = 8$ per group). (**e**) Lipid absorption was determined on 3 consecutive days using sucrose polybehenate (5% of dietary fat, w/w) as internal standard in the HFD. Fat absorption is calculated from the ratios of behenic acid to other FA in diet and faeces, analysed by gas chromatography of FA methyl esters ($n = 9$ per group). (**f**) Lipid tolerance tests were performed by oral gavage of 200 μl olive oil and subsequent determination of acylglycerol levels in plasma of control and ATGLi mice ($n = 5$ per group). (**g**) Heparin releasable LPL activity was determined in WAT of re-fed animals after 50 days of diet intervention ($n = 5$ per group). (**h**) mRNA expression of PPARγ-target genes was measured in gonadal WAT of re-fed mice ($n = 5$ per group). Data represent mean ± s.d. Statistical significance between control and ATGLi was determined by two-tailed Student's *t*-test; *$P < 0.05$, **$P < 0.01$ and ***$P < 0.001$.

**Atglistatin treatment protects mice from HFD-induced NAFLD.** A crucial question of our study addressed the effect of Atglistatin on ectopic lipid accumulation. Humans and mice globally lacking *Atgl* present with systemic lipid accumulation in numerous non-adipose tissues. TG accretion in *Atgl* deficiency is most pronounced in tissues with high oxidation rates such as the liver, heart, BAT and kidney, and the complete lack of the enzyme is incompatible with normal tissue/organ function[5,13]. Notably, patients with increased WAT lipolysis due to mutations in perilipin-1 also develop severe hepatic steatosis (and presumably muscle steatosis) as a result of increased FA delivery from WAT to the liver and other ectopic tissues[9,10]. This indicates that Atgl inhibition specifically in WAT may protect HFD-fed mice from excessive ectopic lipid accumulation. To test this hypothesis, mice received a HFD for 50 days followed by 140 days of HFD feeding in the presence or absence of

Atglistatin. Mass spectrometric analysis of tissue extracts detected the highest accumulation of the inhibitor in WAT, BAT and liver. Although a range of 500 pmol g$^{-1}$ tissue seems low, we assume that the highly hydrophobic structure of Atglistatin (logP = 2.85) causes a much higher local concentration on lipid droplets. The inhibitor was barely detectable in other tissues such as the muscle or heart (Fig. 5a), confirming our previous results with a single Atglistatin gavage[18].

In accordance with the inability of Atglistatin to target skeletal or cardiac muscle, we did not observe increased TG accumulation in these tissues. In fact, even after a long-term treatment of 140 days, the TG content in the skeletal muscle and heart was reduced by 53% and 48%, respectively, in HFD-fed ATGLi compared with control mice (Fig. 5b). As decreased PPARα/PGC1α signalling and defective mitochondrial function leads to lethal cardiac myopathy in *Atgl*-deficient mice[14], we also assessed

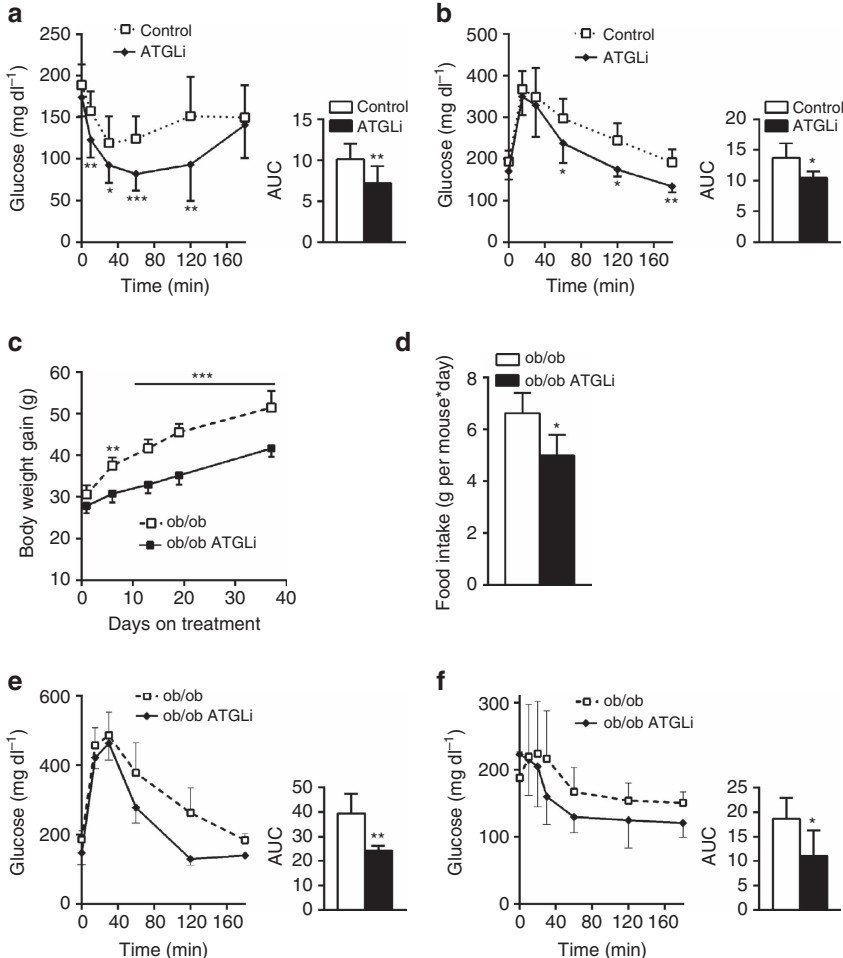

**Figure 4 | Atglistatin-mediated inhibition of lipolysis improves glucose homeostasis. (a,b)** Six weeks old male C57Bl6J mice were fed a HFD (45 kJ% fat; 22.1 kJ g$^{-1}$) for 50 days. Thereafter, mice were fed a HFD in the presence and absence of Atglistatin. **(c–f)** Twenty weeks old *ob/ob* mice were fed a chow diet in the presence and absence of Atglistatin. **(a,f)** Insulin sensitivity and **(b,e)** glucose tolerance was determined after 30 and 40 days diet intervention by i.p. injection of 0.5 IU kg$^{-1}$ insulin and 1.5 g kg$^{-1}$ glucose, respectively ($n = 6$). Area under the curve (AUC) was calculated using GraphPad Prism software. **(c)** Body weight of *ob/ob* animals fed a chow diet in the presence and absence of Atglistatin ($n = 6$). **(d)** Averaged daily food intake of *ob/ob* animals fed a chow diet in the presence and absence of Atglistatin ($n = 6$). Data represent mean ± s.d. Statistical significance was determined by Student's two-tailed *t*-test; *$P < 0.05$, **$P < 0.01$ and ***$P < 0.001$ for control versus ATGLi.

parameters of mitochondrial and cardiac function in ATGLi mice. Cardiac muscle mRNA levels for Pparα, Cpt1β and Pgc1α were similar or even slightly higher in ATGLi mice than in control animals (Fig. 5c). Moreover, cardiac mRNA levels for the fibrosis marker Col1a1 and the macrophage-specific marker F4/80 were reduced by 37% and 32%, respectively, in ATGLi mice (Fig. 5d). In functional terms, left ventricular heart mass and ejection fraction were similar in ATGLi compared with control mice (Fig. 5e,f). Thus, the pharmacological inhibition of Atgl by Atglistatin does not impact cardiac TG metabolism or heart function.

In the liver, total hepatic acylglycerol (TGs, diacylglycerols and monoglycerides) content was 75% lower in HFD-fed ATGLi mice than in HFD-fed control mice (Fig. 5g). To test whether this substantial anti-steatotic effect of Atglistatin can be phenocopied in a genetic model of fatty liver disease, we treated (chow-fed) WT and *ob/ob* mice with the Atgl inhibitor. Similar as observed in the HFD model, Atglistatin treatment caused a 50% and 73% reduction of the hepatic acylglycerol content in WT and *ob/ob* animals, respectively (Fig. 5h). The lipid-lowering effect of Atglistatin is target specific, because Atglistatin had no effect on the increased hepatic fat content in AKO/cTg mice that lack

Atgl in all tissues, except cardiac muscle (Supplementary Fig. 8). The contrary results on liver steatosis in ATGLi mice and global as well as liver-specific *Atgl* knockout (AKO) models was unexpected and indicates that Atglistatin does not inhibit hepatocyte lipolysis sufficiently (unlike genetic inactivation) to cause TG accumulation. This conclusion is supported by identical TG hydrolase activities and slightly increased Atgl protein levels in liver extracts of inhibitor-treated and -untreated mice (Supplementary Fig. 9a,b). We assume that the transient nature of Atglistatin's action and/or the compensatory activity of alternative, currently unknown, TG hydrolases maintain sufficient lipolytic capacity in the liver to prevent the hepatosteatosis observed in mice and humans that totally lack Atgl. Concomitantly, the inhibition of Atgl in WAT reduces FA transport to the liver resulting in decreased fatty liver development in HFD-fed or *ob/ob* ATGLi mice. Consistent with this reduced FA flux from WAT to the liver, ATGLi animals exhibited reduced expression of genes involved in lipid uptake, lipid storage, lipid oxidation and *de novo* lipogenesis (*Pparα* −42%, *Pparγ1* −55%, *Srebp1c* −36%, *Lpl* −50% and *Cd36* −78%), as well as genes driving gluconeogenesis (*Pepck* −30% and glucose 6 phosphatase:

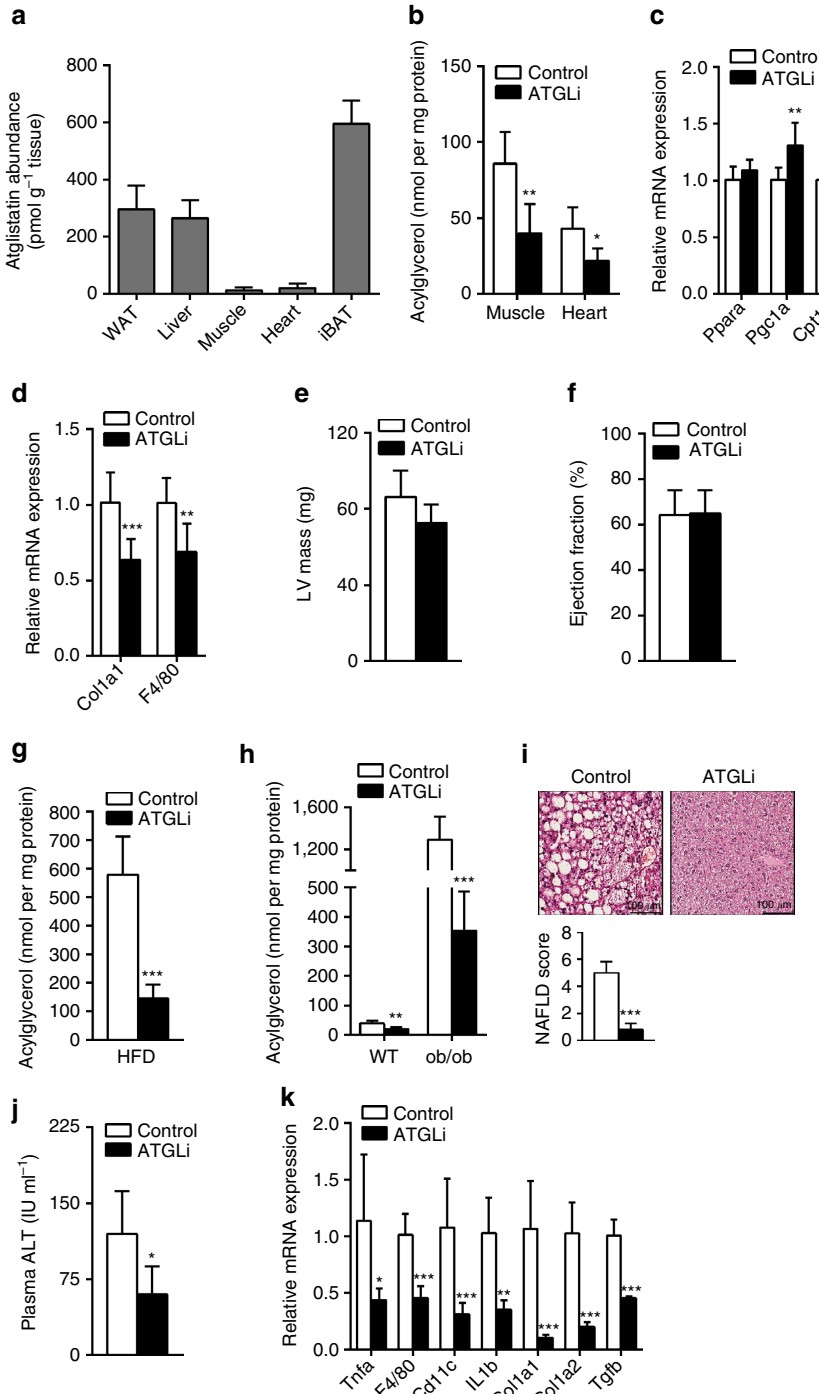

**Figure 5 | Atglistatin protects from HFD-induced NAFLD.** Six weeks old male C57Bl6J mice were fed a HFD (45 kJ% fat; 22.1 kJ g$^{-1}$) for 50 days. Thereafter, mice were fed a HFD in the presence and absence of Atglistatin for 140 days. (**a**) Atglistatin abundance was determined in extracts of adipose- and non-adipose tissues using liquid chromatography–mass spectrometry analysis ($n = 6$ per group). (**b,g,h**) Total lipids were extracted and acylglycerol levels were determined in (**b**) m. quadriceps, and heart and (**g**) livers of wt mice fed a HFD or (**h**) in livers of *ob/ob* mice fed a chow diet in the presence or absence of Atglistatin ($n = 6$ per group). (**c**) mRNA expression of Pparα, Cpt1b and Pgc-1α was assessed in hearts of moderately fasted animals ($n = 6$ per group). (**d**) mRNA expression of inflammatory and fibrosis marker genes was assessed in hearts of re-fed animals ($n = 9$ per group). (**e**) Left ventricular mass and (**f**) ejection fraction of hearts from control and ATGLi animals was assessed by MRI after 5 weeks of diet intervention ($n = 5$ per group). (**i**) Haematoxylin–eosin sections of liver tissues were categorized according to lipid droplet abundance, inflammatory foci and cell ballooning. NAFLD score is the sum of the steatosis, inflammation and ballooning score. (**j**) ALT enzyme activity was measured in fresh isolated plasma using a commercial kit ($n = 9$ per group). (**k**) mRNA expression of inflammatory and fibrosis marker genes was assessed in livers of re-fed animals ($n = 7$ per group). Data represent mean + s.d. Statistical significance between control and ATGLi was determined by two-tailed Student's *t*-test; *$P < 0.05$, **$P < 0.01$ and ***$P < 0.001$.

− 61%) (Supplementary Fig. 9c). Interestingly, the expression of *G0s2*, an Atgl inhibitor and crucial mediator of steatosis in the liver[21] was also sharply reduced on both mRNA (− 75%) and protein level (− 86%) (Supplementary Fig. 9c,d). Finally, hepatoteatosis-associated histological markers typical for NAFLD (inflammation, fibrosis and ballooning) were drastically reduced in ATGLi mice. ATGLi mice displayed lower total NAFLD score (− 84%) (Fig. 5i), reduced plasma alanine aminotransferase (ALT) activity (− 50%) and decreased mRNA expression levels for both inflammatory markers (Tnfα − 62%, F4/80 − 55%, Cd11c − 71% and Il1β − 65%) and markers for hepatic fibrosis (Col1a1 − 90%, Col1a2 − 80% and Tgfβ − 54%) (Fig. 5j,k).

## Discussion

High plasma FA concentrations are a well-established risk factor for the development of IR, type-2 diabetes and NAFLD. Although the mechanistic basis of this association is not entirely clear, inhibition of FA mobilization in WAT represents a rational approach to lower plasma FA concentrations and prevent the development of the aforementioned metabolic disorders. Our current study tested this concept in mice with HFD- and genetically induced obesity, IR and liver steatosis. Chronic administration of Atglistatin, a competitive small molecule inhibitor of the major TG hydrolase Atgl, led to diminished FA release from adipose tissue and decreased plasma FA concentrations. Atglistatin effectively inhibits murine Atgl but fails to inhibit the human enzyme. We find this remarkable considering the structural similarities of mouse and human Atgl in the patatin domain (84% amino acid identity) and warrants more detailed structure function analysis.

Atglistatin-mediated inhibition of lipolysis was associated with an astonishingly beneficial metabolic phenotype. HFD-fed mice treated with Atglistatin were leaner than untreated HFD-fed mice, remained highly insulin sensitive and were resistant to the development of NAFLD. Atglistatin treatment, even when extended to a period of 5 months, did not cause systemic TG accumulation in ectopic tissues such as the skeletal muscle, cardiac muscle or liver. In fact, all of these tissues exhibited lower TG content compared with untreated mice. Importantly, heart function was not impaired in ATGLi mice. This finding contrasts with observations in AKO mice[5,22] and humans with neutral lipid storage disease with myopathy[13], where functional enzyme deficiency leads to systemic TG accumulation, tissue and organ dysfunction, and premature death due to cardiomyopathy. Two points may explain this difference between genetic and pharmacological models of Atgl deficiency. First, Atglistatin is a competitive and reversible inhibitor. We show that 8 h after drug administration, the compound has lost its ability to inhibit lipolysis *in vivo*. Accordingly, depending on the feeding behaviour, mice will undergo a circadian recovery of lipolysis when they are fasting. Complete absence of Atgl activity for 24 h a day—as it happens in AKO mice—is unlikely to occur when Atglistatin is administered via food supply. Second, the tissue distribution of Atglistatin is very uneven. Both single[18] and chronic applications resulted in predominant uptake of the inhibitor in adipose tissue and the liver. By contrast, Atglistatin was essentially undetectable in skeletal muscle, cardiac muscle, brain and several other ectopic tissues analysed, suggesting that these organs do not accumulate a pharmacological relevant concentration of the compound.

ATGLi mice are resistant to HFD-induced obesity, exhibit lower plasma leptin and elevated adiponectin levels, and show reduced inflammation in WAT when compared with untreated control mice on HFD. Although it seems counterintuitive that

inhibition of lipolysis in WAT would lead to reduced WAT mass in HFD fed animals, this finding confirms similar observations in 'heart rescued' and adipose-specific AKO mice on HFD[16,17]. We recently showed that these mice accrue less WAT than HFD-fed control mice due to reduced food uptake and a PPARγ-mediated downregulation of lipid synthesis[16]. Analogously, ATGLi mice on HFD also ate less than HFD-fed control mice. However, pair-feeding experiments demonstrated that reduced food intake does not fully account for reduced adipose mass in HFD-fed ATGLi mice. The actual mechanism by which reduced lipolysis regulates feeding behaviour is currently unknown and awaits clarification. Similarly, as in 'heart-rescued' AKO mice, also ATGLi mice synthesize less TG in WAT than control mice, which is likely to be due to a downregulated expression of PPARγ target genes such as *Lpl*, *Cd36* and *Dgat2*. As these enzymes are rate limiting for both the delivery of lipoprotein-associated FAs to WAT and their re-esterification into TG, ATGLi mice exhibited delayed clearance of plasma TG in dietary fat tolerance tests. Finally, HFD-induced WAT inflammation decreased significantly in ATGLi mice compared with untreated mice. This finding is consistent with previous observations that reduced lipolysis causes decreased macrophage infiltration in WAT[23] and reduced Il-6 production[24].

Analyses of intestinal food absorption using bomb calorimetric analysis of faeces revealed that HFD-fed ATGLi mice consistently extracted fewer calories from food than HFD-fed control mice. Although the difference was small, it was significant and can partially explain the lean phenotype of ATGLi mice on HFD. Whether inhibition of intestinal Atgl by Atglistatin contributes to reduced intestinal absorption efficiency remains to be determined. Nonetheless, reduced food intake and reduced intestinal calorie absorption are insufficient to fully account for the decreased fat mass observed in HFD-fed ATGLi mice. A contribution of increased EE to the lean phenotype in ATGLi mice is thus implied. However, locomotor activity, core body temperature and whole-body EE were not different in ATGLi and control mice. Furthermore, we can exclude that thermogenic activity of BAT contributes to reduced weight gain in ATGLi mice. Although we were able to trace the metabolic fate of all ingested calories in control animals, we could not recover 10% of ingested calories in ATGLi mice. We propose that the difference is caused by increased EE, yet indirect calorimetry is not sensitive enough to detect subtle changes occurring over an extended time period[25]. Taken together, our findings support the concept that a complex cross communication exists between lipolysis and appetite regulation, intestinal calorie absorption and lipid synthesis in WAT. This adaptive interdependence between lipolysis and lipid deposition in WAT may lead to the observed resistance to HFD-induced obesity in both pharmacological and genetic models of Atgl inhibition.

Intriguingly, HFD-fed ATGLi mice were much more glucose tolerant and insulin sensitive than control mice on a HFD. High insulin sensitivity in ATGLi mice is maintained despite decreased plasma insulin levels in ATGLi mice. These data are consistent with findings in global and adipose-specific AKO mice and highlight the importance of FA supply from WAT for the pathogenesis of the disease[16,17,22]. Interestingly, also overexpression of Atgl in adipose tissues attenuates diet-induced obesity and improves glucose homeostasis, indicating that reduced, as well as increased Atgl activity positively influence glucose homeostasis, although via different mechanisms[26]. According to a recent study by Perry *et al.*[27], reduced FA release from WAT of adipose-specific AKO mice limits hepatic acetyl-CoA concentrations and lowers gluconeogenesis, leading to improved whole body glucose homeostasis. Consistent with this concept, ATGLi mice exhibited reduced expression of genes

involved in hepatic gluconeogenesis and impaired gluco-neogenesis after an insulin challenge. Whether a similar direct relationship between lipolysis and IR also exists in humans is less clear. Savage and colleagues[9,10] clearly demonstrated that the unrestrained basal lipolysis in WAT observed in patients affected with mutations in perilipin-1 leads to severe IR and the development of type-2 diabetes arguing for a crucial role of FA in the pathogenesis of the disease. On the other hand, limited data available on glucose tolerance and insulin sensitivity in patients lacking ATGL provide no evidence that lipase deficiency improves insulin sensitivity. Two reported cases with HSL deficiency were even more insulin resistant than normal controls[28].

The most unexpected finding of this study was the robust effect of Atglistatin on liver pathology. The compound substantially reduced HFD-induced hepatosteatosis, liver inflammation and hepatic fibrosis. Compared with HFD-fed control mice, ATGLi mice had 75% less liver fat. A similar anti-steatotic effect of Atglistatin was observed in *ob/ob* mice, a genetic model of obesity, IR and NAFLD. Liver fat was reduced by more than 70% when *ob/ob* mice were treated with the Atgl inhibitor for 6 weeks. Even chow diet-fed WT mice displayed 50% reduced liver fat content. These findings suggest that WAT-derived FAs drive hepatic fat accumulation and that interference with FA transport from WAT to the liver can halt this process. This beneficial phenotype in ATGLi mice is diametrically different from the hepatic phenotype observed in various genetic mouse models lacking Atgl or its coactivator Cgi-58 in the liver. In these models, all animals developed liver steatosis to a varying extent[5,29,30]. Instead, pharmacological inhibition of Atgl resembles the phenotype of adipose tissue specific *Atgl*[17,27] and *Cgi-58* knockout mice[31], suggesting that ATGLi mice retain sufficient Atgl activity in the liver to evade hepatosteatosis. The inability of Atglistatin to inhibit hepatic Atgl and lipolysis may result from increased Atgl enzyme protein levels, the transient nature of Atgilstatin's action or the compensatory activity of alternative, currently unknown, TG hydrolases. The actual mechanisms by which decreased lipolysis in WAT reduces fat accumulation in the liver may include reduced substrate delivery to the liver, increased insulin sensitivity affecting FA utilization[32] and/or decreased expression of hepatic genes known to be regulated by FA sensitive nuclear receptors[33]. Consistent with the latter possibility, we found that the expression of several pro-steatotic genes was drastically reduced in ATGLi mice (*Lpl, Cd36, G0s2* and *Srebp1c*), which may contribute to resistance to hepatic fat accumulation in HFD-fed ATGLi mice.

In conclusion, our study provides profound evidence that transient inhibition of Atgl-driven lipolysis by the selective inhibitor Atglistatin corrects diet induced obesity, glucose intolerance, and fatty liver disease without causing ectopic lipid accumulation. However, Atglistatin does not inhibit lipolysis in human adipocytes. Hence, the development of pharmacological inhibitors targeting human ATGL in WAT may be a useful strategy to combat obesity and obesity related disorders such as IR and NAFLD.

## Methods

**Ethical approval.** All animal studies were approved by and performed according to the guidelines of the Ethics committee of the University of Graz, the Austrian Federal Ministry for Science and Research, and are in accordance with the council of Europe Convention (ETS 123).

**Animals.** Heart rescued *Atgl*-deficient mice were bred as described[14,15]. *Ob/ob* (Lep[ob]/Lep[ob]; B6.V-Lepob/JRj) mice were obtained from Janvier labs. All studies were conducted in male mice on a C57Bl6J background (for at least ten generations). Animals were maintained on a regular light–dark cycle (14 h light–10 h dark) at $22 \pm 1\,°C$ in a specific pathogen-free environment and were *ad libitum* fed a standard laboratory chow diet (M-Z extrudate, V1126,

Ssniff Spezialdiäten, 4.5% fat, 34% starch, 5.0% sugar and 22.0% protein) or HFD (EF R/M D12451 mod., Ssniff Spezialdiäten, 23.1% fat, 8.6% starch, 29.4% sugar and 22.5% protein), except when otherwise stated. HFD intervention studies were started at the age of 6 weeks and continued for 15 weeks or 28 weeks. For pharmacological inhibition of Atgl, mice were randomly selected to be fed a HFD or chow diet supplemented with $2\,\text{mmol}\,\text{kg}^{-1}$ Atglistatin[18]. Before Atglistatin-HFD treatment, mice were fed a HFD without Atglistatin for 50 days unless otherwise stated.

For pair-feeding experiments, mice were single-housed and food for control animals was provided daily according to the food intake of ATGLi mice. For physiological fasting experiments, mice were fasted from 07:00 to 12:00 h, except when otherwise stated. For re-fed status, mice were fasted from 00:00 h until 07:00 h and then re-fed for 2 h.

**Thermoneutrality.** For thermoneutrality studies, mice were fed a HFD for 14 weeks at an ambient temperature of 21–23 °C. Before ATGLi HFD diet intervention was started, mice were acclimatized to thermoneutrality at 30 °C for further 2 weeks. Diet intervention using ATGLi HFD or control HFD was continued at the new temperature for 30 days. Mice were supplied with fresh food on a daily basis. After 21 days *ad libitum* feeding, the control HFD group was switched to pair-feeding regimen receiving the mean daily food intake of ATGLi HFD group from day 0 onwards. The pair-fed group received ~1/3 of total daily food in the morning at ~09:00 h and the residual food between 17:00 and 18:00 h.

**Atglistatin.** Atglistatin is a registered Trademark of the University of Graz and protected by US patent 9,206,115. Atglistatin was synthesized as previously described[18]. Atglistatin containing cookies were prepared by adding $2\,\text{mmol}\,\text{kg}^{-1}$ Atglistatin powder to powderized HFD or chow diet.

**Tolerance tests.** Insulin and glucose tolerance were monitored in awake mice that were fasted for 4 h (08:00–12:00 h) and 5 h (08:00–13:00 h), respectively. For i.p. ITT, mice received an i.p. injection of 0.5 IU insulin per kilogram of body weight. For i.p. GTT, mice received an i.p. injection of 1.5 g glucose per kilogram of body weight. Blood was taken by tail vein puncture and glucose levels were determined at the indicated time points using Wellion Calla classic (Med Trust Holding Ges.m.b.H). Lipid tolerance was determined in mice that were fasted for 4 h (07:00–11:00 h) before they received a gavage containing 200 µl of olive oil. Blood was taken by tail vein puncture and plasma TG levels were determined at the indicated time points using TG Infinity Reagent (Thermo Scientific).

**Body composition and energy balance.** Body mass composition of non-anaesthetized mice was assessed using the time-domain NMR minispec (Live Mice Analyzer system, Model LF90II, Bruker Optik). For cumulative food intake, food consumption was measured every second day on single housed mice. Faeces output was measured on 3 consecutive days using litter-free cages. Locomotor activity, oxygen consumption and carbon dioxide production of animals were monitored by using a laboratory animal monitoring system (PhenoMaster, TSE Systems). Before metabolic phenotyping on four consecutive days, mice were familiarized to single housing and drinking flasks for at least 48 h. Data of the first recorded 24 h were excluded from all analyses. Data were separated on the basis of light–dark cycle, averaged over three light–dark cycles, and are presented as means over animal groups during light–dark period. For analyses of core body temperature, mice were implanted with a telemetry device TA-F10 (Data Sciences International). Therefore, mice were anaesthetized with 80 µg of ketamine and 8 µg of xylacin per gram of body weight. The abdominal region was depilated and prepared with betadine. A small vertical incision was made in the center of the abdomen, the peritoneal cavity was opened and a sterile telemetry device was inserted. The peritoneum and the abdomen were closed by using absorbable sutures and wound clips, respectively. Mice were kept on heating plates until full consciousness was retrieved and received 0.1 mg of enrofloxacin and 2 mg of ibuprofen per millilitre of drinking water for 1 week. Mice were allowed to recover from surgery for at least 2 weeks before wound clips were removed and analyses was performed. core body temperature (CBT) was assessed at an interval of 2 min for three consecutive days.

**Faecal analysis and lipid absorption.** Faeces of mice were sampled on 3 consecutive days and analysed for the excreted energy using a bomb calorimeter. For analysis, 0.7 g faeces were grounded and pressed. The pellets were burned in an adiabatic oxygen bomb calorimeter C4000 A (IKA Analysentechnik, Germany). For lipid absorption determination a HFD containing sucrose polybehenate (5% of total dietary fat content; w/w) was prepared and fed the mice for 3 consecutive days. Fresh faeces were collected twice a day (09:00 and 21:00 h), extracted and analysed by gas chromatography of FA methyl esters as described[20]. In brief, 10 mg faeces and diet were extracted using chloroform/methanol (2/1), 1% acetic acid, 500 µM butylated hydroxytoluene (BHT) for 1 h at room temperature. The aqueous phase was reextracted using chloroform and 500 µM BHT for 15 min at room temperature (RT). The combined organic phases were evaporated under a stream of nitrogen and FA methylester where generated from

whole lipid extracts using methanolic hydrogen chloride[34]. Samples were reconstituted in 100 µl hexane and measured by gas chromatography-flame ionization detector (GC-FID). A wall-coated fused silica 25 m, 0.32 mm ID column (FFAB-CB for free FAs, Varian) was used. One microlitre of the sample was injected at an injector temperature of 230 °C; column A: temperature gradient from 150 °C (hold for 0 min) to 250 °C (hold for 2 min) with 5 °C min$^{-1}$ and a second ramp to increase the temperature to 260 °C (hold for 5 min) with 10 °C min$^{-1}$ was applied. Detector settings: Base temperature, 200 °C; ignition threshold, 0.2 pA; air flow rate, 200 ml min$^{-1}$; H$_2$ flow rate, 30 ml min$^{-1}$; makeup, 20 ml min$^{-1}$. The absorption of FAs was calculated from the ratios of behenic acid to other FA in diet and feces. Investigators were blinded during analysis.

**Atglistatin measurement.** Atglistatin of tissue preparations ($>10$ mg) was extracted twice according to Folch et al.[35] using chloroform/methanol/water (2/1/0.6, v/v/v) containing 500 nmol l$^{-1}$ BHT and 15 pmol internal standard (NM-421, Atglistatin derivative) per sample. Extraction was performed under constant shaking for 60 min at RT. After centrifugation at 1,000 g for 15 min at RT the organic phase was collected. Combined organic phases of the double-extraction were dried under a stream of nitrogen and dissolved in chloroform/methanol/2-propanol (2/1/12, v/v/v) for liquid chromatography–mass spectrometry analysis. Chromatographic separation was performed using an Advance-UHPLC system (Bruker, Billerica, Massachusetts, USA), equipped with a Kinetex C18 column (2.1 × 50 mm, 1.7 µm; Phenomenex, Torrance, California, USA). Solvent A and B consisted of methanol/water (1/1, v/v) and 2-propanol, respectively, containing 0.1% formic acid and 10 mmol l$^{-1}$ ammonium acetate. An EVOQ Elite mass spectrometer (Bruker) equipped with an electro spray ionisation (ESI) source was used for detection. Analyte ions were monitored in multiple reaction monitoring mode (Atglistatin, Qualifier 284→239, Quantifier 284→224; NM-421, Qualifier 270→255, Quantifier 270→227). Atglistatin from 25 pmol l$^{-1}$ to 250 nmol l$^{-1}$ was used for calibration.

**Blood chemistry.** Blood glucose and lactate was determined using Wellion Calla classic (Med Trust Holding Ges.m.b.H) and Accutrend Plus (cobas, Roche Diagnostics), respectively. Plasma levels of cholesterol, FA, TG, glycerol and ketone bodies were analysed using CHOL (Roche Diagnostics), NEFAC (WAKO Chemicals), TG Infinity Reagent (Thermo Scientific), Free Glycerol Reagent (Sigma Aldrich) and Beta-Hydroxybutyrate Assay Kit (Cayman), respectively. Plasma insulin, leptin, adiponectin and IL-6 were determined by using mouse insulin (90080, Chrystal Chem), mouse leptin (90030, Chrystal Chem), mouse adiponectin (K1002-1, B-Bridge International, Inc.) and mouse IL-6 (88-7064, Affymetrix eBiosciences) ELISA kits. QUICK index (quantitative insulin sensitivity check index) was calculated using the formula: 1/(log(fasting insulin µU ml$^{-1}$) + log(fasting glucose mg dl$^{-1}$)).

**Histological analysis.** Tissues were resected, inflated and fixed with 4% neutral buffered formalin. After paraffin embedding (Tissue Tek Tec, Sakura), samples were sectioned (2 µm) and stained with haematoxylin–eosin according to standard histo-pathological techniques. To evaluate the degree of lipid accumulation (steatosis score), tissues were categorized into four grades as follows: no lipid droplets (score = 0); lipid droplets in $<33\%$ of hepatocytes/enterocytes (score = 1); lipid droplets 33–66% of hepatocytes/enterocytes (score = 2); and lipid droplets in $>66\%$ of hepatocytes/enterocytes (score = 3). We counted the number of positive inflammatory cells in ten randomly selected fields per liver section (original magnification: × 400). The liver sections were then classified into four (inflammation score) grades, as follows: no inflammation (score = 0); $<10$ inflammatory foci, each consisting of $>5$ inflammatory cells (score = 1); $\geq10$ inflammatory foci (score = 2) or uncountable diffuse or fused inflammatory foci (score = 3). Furthermore, the degree of liver cell ballooning injury (ballooning score) was classified into three grades as follows: none (score = 0), few balloon cells (score = 1) or many balloon cells/prominent ballooning (score = 2). To evaluate the degree of visceral obesity or fat metaplasia, we counted the mean adipocytes number in five randomly but representative, selected (original magnification: × 200) per section. We calculated the mean diameter of cells after measuring up to 50 representative adipocytes within selected high-power fields (original magnification: × 200) per section. All histological slides were evaluated by independent observers (including certified pathologist) who were blinded to the physical outcome and other biological and pathological data for each sample. In case of disagreement, a consensus score was determined by a third board-certified pathologist.

**Real-time qPCR.** Total RNA was extracted from murine tissues using TRIzol reagent according to the manufacturer's instruction (Life Technologies). Reverse transcription of RNA was performed using random primers (Life Technologies) and gene expression analyses were performed by qPCR using the CFX96 Real-Time PCR System (BioRad) and SYBR Green (Thermo Scientific) technology. Relative mRNA levels were quantified by by ΔΔCt method and using 36B4 as housekeeping gene. Gene-specific primers used for PCR are listed in Supplementary Table 1.

**Tissue preparation and western blot analysis.** Tissues were excised, extensively washed in cold 1 × PBS and homogenized in buffer A (250 mM sucrose, 1 mM dithiothreitol, 1 mM EDTA, 1 µg ml$^{-1}$ pepstatin, 2 µg ml$^{-1}$ antipain, 20 µg ml$^{-1}$ leupeptin) on ice using an Ultra Turrax (IKA, Staufen, Germany). Homogenates were centrifuged for 15 min at 1,000 g, 4 °C and the infranatant was collected. Protein concentration was determined using Bio-Rad Protein Assay and BSA as standard. Proteins of tissue lysates were separated by SDS–PAGE and blotted onto a polyvinylidene difluoride membrane (Karl Roth, Karlsruhe, Germany). Specific proteins were detected using rabbit polyclonal anti-G0s2 antibody (1:1,000, kindly gifted from Liu Jun, Mayo Clinic[36]), rabbit anti-Atgl antibody (1:2,000, 2138, Cell Signaling), rabbit anti-Hsl (1:5,000, 4107, Cell Signaling), rabbit anti-Ucp1 antibody (1:40,000, ab10983, Abcam), rabbit anti-Ndufs1 (1:50,000, ab157221, Abcam) and rabbit anti-Gapdh antibody (1:15,000, 2118, Cell Signaling), and the respective horseradish peroxidase-coupled secondary antibodies (GE Healthcare). Signal densities were analysed using Bio-Rad Chemidoc MP System Software. Uncropped western blotting images are displayed in Supplementary Fig. 10.

**Plasma ALT activity.** ALT activity was determined in plasma of mice using the Infinity ALT(GPT) Liquid stable reagent according to the manufacturer's protocol (Thermo Fisher Scientific). In brief, 20 µl of plasma were incubated with 200 µl of ALT reagent for a total time of 10 min at 37 °C in a Beckman DU640 spectrophotometer. The reaction is monitored by measuring the rate of the decrease in absorbance at 340 nm min$^{-1}$.

**TG hydrolase activity.** For the determination of TG hydrolase activity, lysates of Cos-7 cells overexpressing human and murine ATGL and CGI-58, respectively, or mouse tissue extracts in a total volume of 100 µl buffer A, were incubated with 100 µl substrate and different concentrations of Atglistatin in a water bath at 37 °C for 60 min. As a control, incubations under identical conditions were performed in buffer A alone. After incubation, the reaction was terminated by adding 3.25 ml of methanol/chloroform/heptane (10/9/7; vol/vol/vol) and 1 ml of 0.1 M potassium-carbonate/0.1 M boric acid (pH 10.5). The mixture was intensively vortexed and centrifuged at 800 g for 10 min, 0.2 ml of the upper aqueous phase was collected and the radioactivity was measured by liquid scintillation counting (Tri-Carb 2100TR). TG substrate was prepared by emulsifying 330 µM triolein (40,000 c.p.m. nmol$^{-1}$ glycerol tri[9,10(n)-3H]-oleate (PerkinElmer)) and 45 µM phosphatidylcholine/phosphatidylinositol (3:1) in 100 mM potassium phosphate buffer (pH 7.0) by sonication and adjusted to 5% essentially FA-free BSA (Sigma, St Louis, MO).

**LPL activity.** Fresh isolated tissues from re-fed mice were weighed, minced with scissors and transferred to ice-cold tubes containing 1 ml of DMEM, 2% (wt/vol) FA-free BSA and 2 IU heparin. Tissues were incubated at 37 °C for 1 h before the mixture was centrifuged and media was collected to determine heparin-releasable LPL activity. The substrate ($1.2 \times 10^6$ c.p.m. glycerol tri[9,10(n)-3H]-oleate (PerkinElmer) and 1.04 µmol triolein per sample. Lipids were evaporated under a stream of nitrogen and emulsified by sonication (Bandelin SONOPLUS) in 0.1 M Tris/HCl (pH 8.6) and 0.1% Triton X-100. Forty microlitres of heat-inactivated human serum and 40 µl of 10% BSA were added to the substrate. To determine LPL activities, 200 µl of substrate were mixed with 100 µl of media (heparin-releasable LPL) and incubated in a water bath at 37 °C for 1 h. The reaction was stopped and analysed as described for TG hydrolase activity.

**Ex vivo lipolysis.** The release of FA and glycerol from adipose tissue organ cultures was measured as previously described[37]. In brief, gonadal adipose tissue was excised and intensively washed in prewarmed 1 × PBS. Adipose tissue pieces (20 mg) were incubated in 200 µl DMEM containing 2% BSA (FA free) in the presence or absence of 20 µM Atglistatin for 2 h at 37 °C, 5% CO$_2$ and 95% humidified atmosphere. Thereafter, fat explants were transferred to 1 ml extraction solution (chloroform/methanol, 2/1) and incubated for 1 h at 37 °C under vigorous shaking. Then, fat explants were transferred to 500 µl lysis solution (NaOH/SDS, 0.3N/0.1%) and incubated over night at 55 °C under vigorous shaking. Protein content was determined using BCA reagent (Pierce) and BSA as standard. FA and glycerol content of the incubation media was measured using NEFA kit (WAKO).

**Lipolysis of cultured cells.** SGBS preadipocytes were provided by Novo Department of Pediatrics and Adolescent Medicine, University of Ulm, Germany. Cells were maintained in DMEM/Nutrient Mix F12 (GIBCO, 1/1, v/v), supplemented with 8 µg ml$^{-1}$ biotin (Sigma-Aldrich), 4 µg ml$^{-1}$ pantothenic acid (Sigma-Aldrich), 10% FCS, 100 IU ml$^{-1}$ penicillin and 100 µg ml$^{-1}$ streptomycin at 37 °C in 5% CO$_2$, 95% humidified atmosphere. For differentiation, SGBS cells were seeded at 30,000 cells per well in six-well plates. Upon confluency, differentiation was induced using DMEM/Nutrient Mix F12 1/1, v/v, 100 IU ml$^{-1}$ penicillin, 100 µg ml$^{-1}$ streptomycin, 8 µg ml$^{-1}$ biotin, 4 µg ml$^{-1}$ pantothenic acid, 0.01 mg ml$^{-1}$ transferrin (Sigma-Aldrich), 1 µM cortisol (Sigma-Aldrich), 200 pM triiodothyronine (Sigma-Aldrich), 20 nM human insulin (Sigma-Aldrich), 0.25 µM dexamethasone (Sigma-Aldrich), 500 µM 3-isobutyl-1-methylxanthine

(Sigma-Aldrich) and 2 μM rosiglitazone (Axxora, San Diego, CA). On day 4 of differentiation, the medium was replaced with an identical differentiation medium. On day 8 and 11 of differentiation, the medium was replaced by DMEM/Nutrient Mix F12 supplemented with 100 IU ml$^{-1}$ penicillin, 100 μg ml$^{-1}$ streptomycin, 8 μg ml$^{-1}$ biotin, 4 μg ml$^{-1}$ pantothenic acid, 0.01 mg ml$^{-1}$ transferrin, 1 μM cortisol, 200 pM triiodotyronine, 20 nM human insulin. SGBS cells were fully differentiated on day 14. 3T3-L1 fibroblasts (CL-173) were obtained from ATCC (Teddington, UK) and cultivated in DMEM containing 4.5 g l$^{-1}$ glucose and L-glutamine (Invitrogen) supplemented with 10% FCS and antibiotics under standard conditions. Two days after confluence, medium was changed to DMEM containing 10 μg ml$^{-1}$ insulin (Sigma-Aldrich), 0.25 μM dexamethasone (Sigma-Aldrich) and 500 μM isobutylmethylxanthine (Sigma-Aldrich). After 3 and 5 days, medium was changed to DMEM containing 10 and 0.05 μg ml$^{-1}$ insulin, respectively. The release of FA from cultured cells was measured as previously described[37]. In brief, cells were preincubated with different concentrations of Atglistatin for 2 h. Then, the medium was replaced by DMEM containing 2% BSA (FA free, Sigma), 10 μM Forskolin and different concentrations of Atglistatin for 1 h. The release of FA in the medium was determined using commercial kits (NEFA C, WAKO). Protein concentration was determined using BCA reagent (Pierce) after extracting total lipids using hexane/isopropanol (3:2) and lysing the cells using 0.3 N NaOH/0.1% SDS. Cell lines were used at a low passage number and tested to be mycoplasma free. Authenticity was verified by morphological analysis and analysis of adipocyte marker gene expression using reverse transcriptase–qPCR.

**Determination of tissue acylglycerol content.** Total lipids were extracted using the method of Folch[35]. After centrifugation, the organic phase was collected, dried under a stream of nitrogen and dissolved in 2% Triton X-100 by sonication. Acylglycerol levels were determined using TG Infinity reagent (Thermo Fisher Scientific) and glycerol as standard. The protein fraction was dried at 70 °C for 1 h and dissolved in SDS/NaOH (0.2%/0.1 N) at 60 °C for 2 h under vigorous shaking. Protein concentration was measured using BCA reagent (Pierce) and BSA as standard.

**Magnetic resonance imaging.** Magnetic resonance imaging (MRI) measurements of the heart were performed with a 7T small animal MRI (Bruker BioSpec, Ettlingen, Germany) equipped with a 660 mT m$^{-1}$ gradient coil. A cryogenic cooled transmit/receive coil was used for signal reception. Mice were anaesthetized with isoflurane (1.5–2.0%) in O2 (1 l min$^{-1}$) and scanned in supine position. Animal body temperature was maintained by a water heated animal bed. electrocardiography (ECG) electrodes were placed on the right and left forepaws, and a balloon pressure sensor was placed on the abdomen for respiratory gating. Cine MR images were acquired using an ECG-triggered and respiratory-gated gradient echo sequence with the following measurement parameters: pulse repetition time/echo time = 11.5/2.65 ms, α = 15°, 2 averages, an image matrix of 192 × 192 at a field of view (FOV) of 25 × 25 mm$^2$ and a slice thickness of 1 mm. Two slices were acquired per repetition with an inter-slice gap of 1 mm. Whole heart coverage in short axis view was achieved by four slice packages interleaved by 1 mm inter-slice shift, resulting in eight image slices without slice gaps covering the heart from apex to base. MRI acquisitions started at the up-slope of the ECG R-wave and ten cardiac frames were consecutively recorded. Triggered total imaging time was ∼6 min. The myocardial wall was manually segmented in all short axis views using itk-SNAP version 3.2.0 to obtain left ventricular (LV) epicardial and endocardial volumes at the end-diastole and end-systole, while papillary muscles were excluded from the lumen[38]. LV mass was calculated from end-diastolic myocardial volume using a tissue density of 1.04 g cm$^{-3}$. Ejection fraction was obtained by the difference of end-diastolic and end-systolic endocardial volume divided by the end-diastolic endocardial volume multiplied by 100%.

**Statistical analyses.** Data are shown as mean ± s.d. Statistical analysis was performed on data distributed in a normal pattern between two groups by Student's two-tailed $t$-test. For analysis of multiple measurements, we performed one-way analysis of variance followed by Bonferroni *post-hoc* test using GraphPad Prism version 6.00 for Windows, GraphPad Software, La Jolla, California, USA. To test for the influence of body weight and body composition variation on EE, group comparisons were adjusted for body weight and fat mass in separate analyses using ANCOVA[39]. ANCOVA was performed with the univariate general linear model module in SPSS statistics. For all analyses, group differences were considered statistically different for *$P < 0.05$, **$P < 0.01$ and ***$P < 0.001$ for comparison between control and ATGLi animals, and #$P < 0.05$,##$P < 0.01$ and ##$P < 0.001$ for comparison between different genotypes and feeding status. Nalimov test was performed to identify outliers if samples had to be excluded from the analysis. Group-size estimations were based upon a power calculation to minimally yield an 80% chance to detect a significant difference of $P < 0.05$ between groups.

**Data availability.** The authors declare that the data supporting the findings of this study are available within the paper and its Supplementary Information files, or are available from the authors upon reasonable request.

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

## Acknowledgements

We thank Astrid Steiner and Birgit Juritsch for animal care and genotyping, and Silvia Schauer and Wael Al-Zoughbi for histological analysis. Financial support was given by the European Research Council under European Union's Seventh Framework Programme Grant FP/2007-2013/ERC Grant Agreement 340896, LipoCheX (to R. Zechner), the Austrian Science Fund (FWF) through the project W901 Doktoratskolleg Molecular Enzymology (to R. Zechner, R. Zimmermann and R.B.), the project P24294 (to R. Zimmermann), SFB LIPTOX F30 (to R. Zechner) and P28286 (to R.B.), the Louis-Jeantet Prize for Medicine (to R. Zechner) and the grant 12CVD04 from the Fondation Leducq (to R. Zechner).

## Author contributions

M.S. conceived the study, designed and performed *in vitro*, *in vivo* and *ex vivo* mouse experiments, analysed and interpreted data, and co-wrote the manuscript. M.R. performed *in vivo* and *in vitro* experiments. R.S. performed *in vitro* experiments, conducted surgery and interpreted data, to evaluate core body temperature, and performed thermoneutrality studies. G.F.G. performed lipid analysis experiments and assisted *in vivo* experiments. S.H. performed mRNA and protein expression analysis and assisted with *in vivo* experiments. P.K. performed chow diet studies. P.B. performed *in vitro* experiments and assisted in mouse studies. T.O.E. and O.K. performed and interpreted HPLC, GC and MS experiments. S.Y. performed histological analysis. W.D.C. assisted for bomb calorimetry measurement. C. Diwoky performed MRI. C. Doler, N.M. and R.B. designed and synthesized Atglistatin. R. Zimmermann designed research, interpreted results and helped writing the manuscript. R. Zechner designed research, interpreted results and co-wrote the manuscript.

## Additional information

**Competing interests:** The authors declare no competing financial interests.

DOI: 10.1038/ncomms15490    OPEN

# Corrigendum: Pharmacological inhibition of adipose triglyceride lipase corrects high-fat diet-induced insulin resistance and hepatosteatosis in mice

Martina Schweiger, Matthias Romauch, Renate Schreiber, Gernot F. Grabner, Sabrina Hütter, Petra Kotzbeck, Pia Benedikt, Thomas O. Eichmann, Sohsuke Yamada, Oskar Knittelfelder, Clemens Diwoky, Carina Doler, Nicole Mayer, Werner De Cecco, Rolf Breinbauer, Robert Zimmermann & Rudolf Zechner

Nature Communications 8:14859 doi: 10.1038/ncomms14859 (2017); Published 22 Mar 2017; Updated 25 Apr 2017

The affiliation details for R. Zimmermann, R. Zechner and R. Breinbauer are incorrect in this Article. The correct affiliation details for these authors are given below:

Institute of Molecular Biosciences, University of Graz, Heinrichstrasse 31, 8010 Graz, Austria.

BioTechMed-Graz, Mozartgasse 12, 8010 Graz, Austria.

