## [Peer Review File · Nature Communications]

Reviewer #1 (Remarks to the Author)

The authors provide an interesting data set concerning the effects of an ATGL inhibitor on energy balance and indices of insulin action in mice fed a high fat diet. The authors show that ATGL inhibition effectively improves the metabolic profile in mice on HFD and ob/ob mice fed chow by substantially ameliorating obesity, insulin resistance, and liver steatosis. While these are interesting and potentially important observations, the mechanisms involved are complex and somewhat obscure. Many aspects of the phenomenon were recently reported in PNAS. One would hope that a novel chemical inhibitor might provide new mechanistic insights regarding effects on food intake, lipid synthesis, TG handling and glucose metabolism. However, the data seem to be mostly descriptive, and the reasons underlying the fascinating differences between genetic and pharmacological inhibition remain unresolved. Nevertheless, the data do establish a proof of principle for ATGL inhibition.

1) Experiments in figure 1 seem to be contrived to show transient effect, but the design is not strictly relevant to chronic dosing and ad libitum feeding. It would seem difficult to deconvolve the effects of compound on food intake, lipid synthesis, fasting lipid mobilization etc. when administered in the food. The appeal to dose timing and tissue partitioning is speculative.

2) Is the effect of compound reversible?

3) Atglistatin and ATGL KO both reduced white fat pad size of mice on HFD. However, the effects in BAT are highly discordant. The effect of compound on BAT morphology is very striking and is suggestive of activation. Did the authors look at indices of BAT activation, like Ucp1, Cidea, Pgc1a, etc?

4) Measuring Atglistatin levels in adipose tissue and liver does not establish the site of action, as is claimed. The analytical method (Folch extraction) indicates that the compound is highly hydrophobic and the ATGL measured will be that which has partitioned into lipid. In addition, nothing is reported on the PK of Atglistatin, and it seems unlikely that levels of pmol/g are physiologically meaningful. Given the diverse effects on food intake, lipid absorption and clearance, etc. it is not clear what effects are primary and secondary to lipolysis inhibition.

5) The experiments do not test the premise regarding FFA release from adipose tissue.

6) The claim that the lipid lowering effect of Atglistatin depends on the expression of ATGL in the liver, is not demonstrated since ATGL levels in liver specifically were not manipulated nor was ATGL activity measured in the liver.

Reviewer #2 (Remarks to the Author)

Schweiger et al. report the effects of treatment with an inhibitor of adipose triglyceride lipase (ATGL) on obesity, liver steatosis and insulin resistance. The work is based on the identification of the first ATGL inhibitor by the authors (Mayer et al. 2013 Nat. Chem. Biol. 9, 785).

The article reports the first characterization of the effects of an ATGL inhibitor on several conditions predisposing to diabetes, cardiovascular complications and liver cirrhosis. The work has been carefully conducted with appropriate techniques.

As the paper is a preclinical proof of concept for the use of ATGL inhibitors, there are however several points that are important to address.

Main points

1. To assess the specificity of ATGL effects, the authors used transgenic mice which express ATGL only in heart and do not die from cardiomyopathy. Using these mice, the authors show that the lower increase of inguinal white adipose tissue weight during high fat diet is due to ATGL.

However, some important questions cannot be addressed using heart-rescued ATGL knockout mice. Adipocyte-specific ATGL knock out mice treated with the ATGL inhibitor could be useful to:

- prove that the control of fat mass is mediated by adipose ATGL inhibition and activation of PPAR gamma through production of endogenous agonist(s).

- demonstrate that the liver phenotype is not due to inhibition of hepatic ATGL.

- determine whether the reduction of adipose inflammation is due to ATGL expression in macrophages.

2. As mentioned by the Authors, the data suggest that the decreased fat mass observed in high fat

diet-fed ATGL inhibitor-treated mice cannot be entirely explained by reduced food intake and intestinal fat absorption. Yet, no increase in energy expenditure could be observed maybe due to the limitations of indirect calorimetry in capturing moderate changes. The ATGL inhibitor accumulates in white adipose tissue and liver but not in other tissues with ATGL expression (Fig. 5a). The abundance of the inhibitor is not reported in brown adipose tissue. Change in cell size in brown fat (as also reported in white fat in Fig. 1) suggests that the inhibitor accumulates in brown adipose tissue where it could modify thermogenic activity. Treatment of high fat-fed mice with the ATGL inhibitor in a thermoneutral environment may help determining whether brown adipose tissue thermogenesis contribute to the decreased body weight gain.

3. During chronic treatments, assessment of ATGL and HSL protein level and enzymatic activity will document whether the ATGL inhibitor has effects on adipose lipase expression.

Other points

1. The concept of inhibition of adipose tissue lipolysis in the treatment of the metabolic syndrome is not a new one. Nicotinic acid, an antilipolytic molecule acting on a GPCR was proposed in the treatment of dyslipidemia 50 years ago. Nicotinic acid and acipimox, a related molecule, have been tested in humans for action on insulin resistance. Several chemical series of inhibitors for another adipose lipase, hormone-sensitive lipase (HSL), have been synthesized with some compounds being tested on metabolic complications. This rationale needs to be mentioned in the introduction section.

2. Dr. Zechner reported at conferences (and maybe published?) that the molecule studied here is a poor inhibitor of human ATGL. This may be mentioned in the therapeutic perspectives of the Discussion section.

3. Fig. 3. LPL and DGAT are assessed as PPAR γ targets. A more complete survey of PPAR γ target genes with known functional PPAR-responsive elements in their promoters may support the conclusion of an inhibition of PPAR γ activity.

4. HOMA-IR is used in Table 1 to assess insulin sensitivity. Use of HOMA-IR is inappropriate in the mouse as the denominator factor of 22.5 used in the equation is human specific. QUICKI index may be used instead.

Reviewer #3 (Remarks to the Author)

ATGL is the rate-limiting enzyme of first step of lipolysis. Increasing adipose tissue lipolysis activity may increase circulating FFA in serum and lead to insulin resistance and NAFLD. Recent work showed that genetically inhibition of lipolysis could improve insulin sensitivity and protect mice from HFD induced obesity. Here, the authors tried to find whether ATGL inhibitor Atglistatin could act similarly as genetically inhibition. They described many beneficial effects from this drug treatment on HFD mice: plasma FFA is reduced; weight gain of WAT and the whole body is blunted; adipocytes inflammation is decreased; insulin sensitivity is improved by GTT and ITT assays. Surprisingly, the drug also protects mice from HFD-induced liver steatosis. Moreover, the drug-treating effect is better than ATGL genetic deficiency in that ATGL deficiency associated cardiac steatosis and cardiomyopathy is not observed in Atglistatin-treated mice. Overall, Atglistatin could indeed protect mice from HFD induced obesity, IR and liver steatosis based on their data. However, there are several major concerns with this manuscript.

1) As their previous works on genetically knockout ATGL mice show increased body weight and WAT weight in chow condition, what will happen if Atglistatin-treated mice are fed in chow condition? Will it behave similarly? As chronic chow feeding in wild type could also lead to NAFLD and IR, could Atglistatin work well in this case?

2) I noticed that the leptin level reducing remarkably (34.6 vs 3.8) in HFD 5h fasted condition which is shown in Table 1 while both control and ATGLi group food intake showed no significant differences. It is really intriguing to know the leptin level in normal condition, as it is not shown in this table. Basically, leptin regulates appetite effectively in WT mice. May Atglistatin affect leptin signaling pathway? This needs to be illustrated.

3) In oral lipid tolerant test, there is a delayed clearance of post-prandial fat concurred with a sharp reduction of LPL activity. LPL activity is very important to the FA import. Lower LPL activity

could result in fewer FA in cells and subsequently the decreased fat content phenotype in adipose tissue, liver and muscle. Since we don't have much information on Atglistatin target specificity, is it possible that Atglistatin also work as LPL inhibitor, besides its inhibition on ATGL? More characters of this drug should be provided to make this clear.

4) The effect of Atglistatin in liver is surprising and intriguing. Since the concentration of Atglistatin is very high in ATGLi liver, the authors should measure the lipolysis activity in liver extracts.

5) In the discussion, the authors want to discuss the paper published in Diabetes, 2009 by Ahmadian et al, which reported that overexpression of ATGL lead to improved insulin sensitivity.

Point to point response to the referees:

Reviewer #1:

We thank the reviewer for the overall positive and constructive criticism.

Many aspects of the phenomenon were recently reported in PNAS. One would hope that a novel chemical inhibitor might provide new mechanistic insights regarding effects on food intake, lipid synthesis, TG handling and glucose metabolism. However, the data seem to be mostly descriptive, and the reasons underlying the fascinating differences between genetic and pharmacological inhibition remain unresolved. Nevertheless, the data do establish a proof of principle for ATGL inhibition.

We agree with the reviewer that we observed several consistencies between the genetic model of ATGL deficiency and the pharmacological inhibition of ATGL. This accordance confirms the efficacy and target specificity of the inhibitor. However, we also report on some important and unexpected differences between the genetic and pharmacological model. In contrast to findings in global ATGL deficiency, transient inhibition of lipolysis by Atglistatin leads to reduced cardiac and hepatic steatosis, reduced WAT inflammation, and no changes in BAT morphology in mice on a HFD. These beneficial effects upon inhibitor treatment highlight in our view the potential of ATGL as a target to treat obesity and its metabolic consequences.

1) Experiments in figure 1 seem to be contrived to show transient effect, but the design is not strictly relevant to chronic dosing and ad libitum feeding. It would seem difficult to deconvolve the effects of compound on food intake, lipid synthesis, fasting lipid mobilization etc. when administered in the food. The appeal to dose timing and tissue partitioning is speculative.

We agree with reviewer that our data does not represent a solid pharmacodynamic study on the inhibitor. However, the transient nature of inhibitor impact is clearly apparent from the fasting experiments presented in Fig. 1. Ex vivo lipolysis assays using fat explants demonstrate that feeding Atglistatin leads to decreased adipose tissue lipolysis whereas fasting of mice for 8 h completely restores FA release. When administered by gavage or by ip. injection, we previously demonstrated that the effect of Atglistatin is transient (Mayer et al., Nat. Chem. Biol., 2013). Twelve hours after a single Atglistatin administration, FA and glycerol levels are similar in treated and untreated animals arguing for a loss of drug efficacy after this time period. We are aware of the fact that ad libitum inhibitor feeding results in a somewhat undefined fluctuation of inhibitor concentrations in blood and tissues and discuss this fact in relation to the observed phenotypes.

2) Is the effect of compound reversible?

Previously we demonstrated that Atglistatin is a competitive inhibitor of ATGL (Mayer et al., Nat. Chem. Biol., 2013). Its effect on plasma levels of lipolytic products (FA and glycerol) is reversed upon withdrawal

of the inhibitor. In the current manuscript, we additionally show that plasma FA levels almost reach control levels already after 5 h fasting. Moreover, using fat explants, we demonstrate that fasting Atglistatin-treated mice reverses its inhibitory effect on adipose tissue lipolysis *ex vivo*. MS analysis revealed that the compound is not detectable in plasma or tissues of mice fasted for more than 5h (data not shown). Thus we assume that Atglistatin is washed out of the body or degraded, which renders its effects reversible.

3) Atglistatin and ATGL KO both reduced white fat pad size of mice on HFD. However, the effects in BAT are highly discordant. The effect of compound on BAT morphology is very striking and is suggestive of activation. Did the authors look at indices of BAT activation, like Ucp1, Cidea, Pgc1a, etc?

The reviewer raises an important point that we addressed with additional experiments displayed in Supplemental Fig. 4. We assessed mRNA levels of Ucp-1, Pgc1a and Cidea in iBAT and found no differences between control and ATGLi animals (Supplemental Fig 4a). Interestingly, UCP1 protein levels were significantly increased in iBAT of ATGLi mice (Supplemental Fig 4 b). Despite this increase in UCP1 protein, telemetry sensors implanted in to the abdominal cavity of mice did not reveal differences in core body temperature (Supplemental Fig 4c). To assess whether differences in thermogenesis over an extended period of time contribute to reduced weight gain, we fed mice a HFD in the presence and absence of Atglistatin under thermoneutral conditions (Supplemental Fig 4d-h). We show that ATGLi animals were resistant to diet induced obesity also under thermoneutral conditions indicating that the obesity resistant phenotype is not caused by alterations in BAT activity.

4) Measuring Atglistatin levels in adipose tissue and liver does not establish the site of action, as is claimed. The analytical method (Folch extraction) indicates that the compound is highly hydrophobic and the ATGL measured will be that which has partitioned into lipid. In addition, nothing is reported on the PK of Atglistatin, and it seems unlikely that levels of pmol/g are physiologically meaningful. Given the diverse effects on food intake, lipid absorption and clearance, etc. it is not clear what effects are primary and secondary to lipolysis inhibition.

The logP value for Atglistatin is 2.85 making it a highly hydrophobic molecule that presumably binds to cellular lipid and membrane fractions. This also explains why Atglistatin can be efficiently extracted from tissues by the Folch method. We agree with the reviewer that a tissue concentration of Atglistatin in the range of 500 pmol/g (Fig. 5a) is low but assume that the local concentration on LDs may be much higher and explain the efficacy of the inhibitor. This is now discussed in the revised version of the manuscript.

With regard to primary and secondary effects of Atglistatin, we conclude that inhibition of ATGL represents the primary effect of the inhibitor and that all other observed effects and phenotypes are secondary to reduced lipolysis (or potential off target effects).

5) The experiments do not test the premise regarding FFA release from adipose tissue.

The fat pad assay performed in the current study is a well-established *in vitro* method to measure the FA release from adipose tissue (Schweiger et al., Meth Enzymol, 2014; Jaworski et al., Nat Med, 2009, Kosteli et al., JCI, 2010; Gironse et al., PlosBiol, 2013; ...).

6) The claim that the lipid lowering effect of Atglistatin depends on the expression of ATGL in the liver, is not demonstrated since ATGL levels in liver specifically were not manipulated nor was ATGL activity measured in the liver.

We agree that the sentence “The lipid lowering effect of Atglistatin depends on the expression of ATGL in the liver, as liver samples of AKO/cTg mice that completely lack ATGL in all tissues except cardiac muscle exhibited increased hepatic fat content that could not be reduced by Atglistatin” is misleading and changed it to: “The lipid lowering effect of Atglistatin is target-specific, because Atglistatin had no effect on the increased hepatic fat content in AKO/cTg mice that lack ATGL in all tissues except cardiac muscle”.

The contrary results on liver steatosis in ATGLi mice and global as well as liver-specific ATGL knockout models indicates that Atglistatin does not inhibit hepatocyte lipolysis sufficiently (unlike genetic inactivation) to cause TAG accumulation. This conclusion is supported by our finding that TG hydrolase activities in liver extracts of inhibitor treated and untreated mice were identical (Suppl. Fig. 9b). This inability of Atglistatin to inhibit hepatic ATGL and lipolysis may result from increased ATGL enzyme protein levels (Suppl. Fig. 9a), the transient nature of Atglistatin’s action (Mayer et al., Nat Chem Biol, 2013), or the compensatory activity of alternative, currently unknown, TG hydrolases. The existence of alternative lipases is corroborated by the relatively small contribution of ATGL to the total lipolytic activity in liver lysates (see attached Fig.1). The various possibilities to explain or findings are discussed in the paper.

Reviewer #2

We thank the reviewer for his/her insightful and constructive comments.

1. To assess the specificity of ATGL effects, the authors used transgenic mice which express ATGL only in heart and do not die from cardiomyopathy. Using these mice, the authors show that the lower increase of inguinal white adipose tissue weight during high fat diet is due to ATGL. However, some important questions cannot be addressed using heart-rescued ATGL knockout mice. Adipocyte-specific ATGL knock out mice treated with the ATGL inhibitor could be useful to:

(i) prove that the control of fat mass is mediated by adipose ATGL inhibition and activation of PPAR gamma through production of endogenous agonist(s)

(ii) demonstrate that the liver phenotype is not due to inhibition of hepatic ATGL.

(iii) determine whether the reduction of adipose inflammation is due to ATGL expression in macrophages.

We agree with the reviewer that experiments with Atglistatin-treated adipose tissue-specific ATGL-deficient mice would additionally validate Atglistatin's target specificity. However, we provide this validation already in global ATGL-deficient mice in the paper and believe that this model is generally more appropriate to delineate ATGL-dependent and ATGL-independent effects of the inhibitor. We believe that the relatively minor additional information gained by a repetition of these experiments in adipose-specific knockout mice does not warrant a considerable further extension of the revision period by several months, which it would take to re-raise the appropriate animals and perform the requested experiments. Moreover, most questions raised by the reviewer can be answered by comparing the phenotypes observed in tissue-specific ATGL deficient animals and Atglistatin-treated mice.

ad (1) Schreiber et al., (PNAS 2015) demonstrated that heart-rescued ATGL ko mice are protected from HFD induced obesity due to reduced adipose PPAR γ activity and fat synthesis as well as decreased calorie intake. Consistent with these findings, Schoiswohl et al. (Endocrinology, 2015) showed that adipose-specific ATGL-deficient animals on a HFD also exhibited reduced weight gain and reduced expression of genes responsible for *de novo* lipogenesis, lipid uptake, and lipid synthesis (PPAR γ target genes) in WAT. Finally, we demonstrate in the current study that in accordance with the knockout models ATGL inhibition by Atglistatin also reduces adipose tissue mass by reduced food intake and reduced expression of selected PPAR γ target genes in WAT (Fig. 3h) of mice on a HFD. Taken together, these data led to our conclusion that the reduced on PPAR γ target gene expression and decreased fat mass in ATGLi mice is due to the inhibition of ATGL in adipose tissue. This conclusion is reminiscent of a similar outcome in cardiac muscle-specific ATGL-deficient mice, where the lack of ATGL leads to defective PPAR α signaling, mitochondrial dysfunction and cardiomyopathy. Evidently, ATGL-mediated lipolysis is a general requirement for effective PPAR function. Whether lipolytic products of ATGL like FAs and DAGs directly act as PPAR ligands or whether more indirect mechanisms (such as FA or DAG conversion into other ligands of PPARs) are responsible for PPAR activation remains to be elucidated.

ad (ii) ATGL gene deletion in the liver in global or liver-specific ATGL knockout mice causes hepatosteatosis and liver inflammation (Haemmerle, G. et al. Science, 2006; Hoy et al., Endocrinology, 2010; Wu et al., Hepatology, 2011; (LIT.) In contrast, Atglistatin treatment ameliorates hepatic steatosis and inflammation in HFD-fed or *ob/ob* mice. Similarly as ATGLi mice, adipose-specific ATGL deficient-animals also exhibit reduced hepatic lipid levels and reduced inflammatory marker gene expression (Schoiswohl et al., Endocrinology, 2015). Therefore we assume that the beneficial effect of Atglistatin treatment on liver pathology is not caused by the inhibition of hepatic lipolysis but instead by an inhibition of adipose tissue lipolysis and a concomitant decrease in FA flux from adipose tissue to the liver. This assumption is corroborated by the inability of Atglistatin to lower TG hydrolase activity in the liver of ATGLi animals (Supplemental Fig. 9b).

Ad (iii) The reviewer raises a very important point. At this stage, we do not know whether Atglistatin inhibits macrophage ATGL and whether this may cause the antiinflammatory phenotype observed in inhibitor-treated animals. The previous finding that ATGL-deficient macrophages exhibit an M2-like, antiinflammatory phenotype (Aflaki et al, CMLS, 2011), however, potentially support this presumption. Also the fact that, unlike ATGLi mice, adipose-specific ATGL-deficient animals have increased WAT inflammation (F480, cd11c, IL6, TNF α) (Schoiswohl et al, Endocrinology, 2015) argues for a potential antiinflammatory role of the inhibitor via ATGL inhibition in adipose tissue resident macrophages. We believe that the role of Atglistatin in macrophages requires a more extended investigation beyond the

scope of the current work. However, we consider the role of macrophage ATGL and its potential inhibition by Atglistatin in the context of tissue inflammation in the discussion of the manuscript.

2. As mentioned by the Authors, the data suggest that the decreased fat mass observed in high fat diet-fed ATGL inhibitor-treated mice cannot be entirely explained by reduced food intake and intestinal fat absorption. Yet, no increase in energy expenditure could be observed maybe due to the limitations of indirect calorimetry in capturing moderate changes. The ATGL inhibitor accumulates in white adipose tissue and liver but not in other tissues with ATGL expression (Fig. 5a). The abundance of the inhibitor is not reported in brown adipose tissue. Change in cell size in brown fat (as also reported in white fat in Fig. 1) suggests that the inhibitor accumulates in brown adipose tissue where it could modify thermogenic activity. Treatment of high fat-fed mice with the ATGL inhibitor in a thermoneutral environment may help determining whether brown adipose tissue thermogenesis contribute to the decreased body weight gain.

We thank the reviewer for raising this important point. In accordance with her/his recommendations, we performed MS analyses and found high concentrations of Atglistatin in iBAT (Fig. 5a). We also measured core body temperature in mice using abdominally implanted telemetric sensors but did not observe significant differences in Atglistatin-treated or untreated mice (Supplemental Fig. 4). Finally, we followed the suggestion of the reviewer to study mice at thermoneutral conditions. To assess weight loss independent of reduced food intake we pair-fed the animals. We found that similar as in ambient conditions Atglistatin-fed animals lost more body weight due to fat loss (iBAT, iWAT, gWAT) than pair-fed control mice also in thermoneutrality. Unlike in the genetic knockout model, brown adipocytes became smaller by Atglistatin treatment in thermoneutrality (Supplemental Fig. 4). Together, these data argue against an increase in BAT activity upon Atglistatin feeding that would contribute to decreased body weight gain.

3. During chronic treatments, assessment of ATGL and HSL protein level and enzymatic activity will document whether the ATGL inhibitor has effects on adipose lipase expression.

As suggested by the reviewer, we performed Western blot analysis and found that Atglistatin treatment caused an increase in ATGL and a slight decrease in HSL protein abundance in gWAT (Supplemental Fig.1). Despite increased cellular ATGL protein levels, adipose tissue explants from ATGLi mice released less FA and glycerol than explants from untreated mice arguing for effective inhibition by Atglistatin (Figure 1).

Other points

1. The concept of inhibition of adipose tissue lipolysis in the treatment of the metabolic syndrome is not a new one. Nicotinic acid, an antilipolytic molecule acting on a GPCR was proposed in the treatment of dyslipidemia 50 years ago. Nicotinic acid and acipimox, a related molecule, have been tested in humans for action on insulin resistance. Several chemical series of inhibitors for another adipose lipase, hormone-

sensitive lipase (HSL), have been synthesized with some compounds being tested on metabolic complications. This rationale needs to be mentioned in the introduction section.

We agree and extended the introduction accordingly by adding : "Nicotinic acid (niacin) has been used in humans to lower plasma lipids by targeting lipolysis via a G-protein coupled receptor. However, the drug has several off-target side effects limiting its acceptance by patients (Karpe et al., Lancet, 2004). Another approach constitutes the inhibition of either of the two main adipose tissue lipases. To date, only small-molecule inhibitors for HSL have been tested in mice and this treatment resulted in improved insulin sensitivity but did not affect body weight, fat mass, and WAT inflammation in mice fed a high caloric diet (Girousse et al. Plos Biol, 2013)."

2. Dr. Zechner reported at conferences (and maybe published?) that the molecule studied here is a poor inhibitor of human ATGL. This may be mentioned in the therapeutic perspectives of the Discussion section.

We include the requested figure (Figure 1 d, e) showing that Atglistatin does not inhibit human ATGL or human adipocyte lipolysis.

3. Fig. 3. LPL and DGAT are assessed as PPARgamma targets. A more complete survey of PPARgamma target genes with known functional PPAR-responsive elements in their promoters may support the conclusion of an inhibition of PPARgamma activity.

In accordance with this recommendation, we now show reduced WAT expression of Angptl4, Cd36, and G0s2 as typical additional target genes for PPAR γ (Fig. 3h).

4. HOMA-IR is used in Table 1 to assess insulin sensitivity. Use of HOMA-IR is inappropriate in the mouse as the denominator factor of 22.5 used in the equation is human specific. QUICKI index may be used instead.

Again an important point that we were happy to address. We calculated **quantitative insulin sensitivity check index** (QUICKI) for our animals and found a highly significant increase in Atglistatin treated compared to control HFD fed mice, verifying improved insulin sensitivity (included in Table1).

Reviewer #3

We thank the reviewer for the thorough review and important input.

1) As their previous works on genetically knockout ATGL mice show increased body weight and WAT weight in chow condition, what will happen if Atglistatin-treated mice are fed in chow condition? Will it

behave similarly? As chronic chow feeding in wild type could also lead to NAFLD and IR, could Atglistatin work well in this case?

We followed the suggestion of the reviewer and performed the chronic feeding of Atglistatin in animals on chow diet. As shown in Supplemental Fig. 7 and Fig. 5h, effects of Atglistatin on body weight gain, fat mass reduction, and reduction in liver fat were similar but less pronounced than in mice on HFD. Consistent with this more moderate effect, we did not observe significant changes in insulin and glucose tolerance or food intake in chow-fed Atglistatin-treated versus untreated mice.

2) I noticed that the leptin level reducing remarkably (34.6 vs 3.8) in HFD 5h fasted condition which is shown in Table 1 while both control and ATGLi group food intake showed no significant differences. It is really intriguing to know the leptin level in normal condition, as it is not shown in this table. Basically, leptin regulates appetite effectively in WT mice. May Atglistatin affect leptin signaling pathway? This needs to be illustrated.

As suggested by the reviewer, we measured leptin concentrations in re-fed conditions and found them also significantly reduced in Atglistatin treated animals (Table 1).

To answer the reviewer's question on leptin signaling, we performed leptin sensitivity assays (attached Fig. 2). We found both groups (control as well as Atglistatin treated animals) to be identically leptin resistant. Hence, despite markedly reduced leptin levels in ATGLi mice, leptin sensitivity is unaltered compared to untreated animals. We also performed qPCR analysis to assess hypothalamic leptin signaling and found, *Agrp* and *NPY* increased; *POMC* and *CARTPT* decreased in ad libitum fed mice (attached Fig. 2). This indicates that ATGLi mice may have increased appetite but still eat less. This counter-intuitive observation adds to the mystery how reduced adipose tissue lipolysis affects food intake, a question that must be thoroughly addressed in future work and cannot be answered by the current study.

3) In oral lipid tolerant test, there is a delayed clearance of post-prandial fat concurred with a sharp reduction of LPL activity. LPL activity is very important to the FA import. Lower LPL activity could result in fewer FA in cells and subsequently the decreased fat content phenotype in adipose tissue, liver and muscle. Since we don't have much information on Atglistatin target specificity, is it possible that Atglistatin also work as LPL inhibitor, besides its inhibition on ATGL? More characters of this drug should be provided to make this clear.

We agree that this is a very important point. In a previous publication (Mayer et al, Nat. Chem. Biol., 2013) we investigated the target specificity of Atglistatin and showed that Atglistatin does not directly affect LPL activity. In this work we measured heparin releasable LPL activity from adipose tissue and found it to be drastically reduced. This reduction is presumably due to reduced LPL expression as a result of reduced PPAR γ signaling and goes along with a downregulation of other PPAR γ target genes (Fig.3h).

4) The effect of Atglistatin in liver is surprising and intriguing. Since the concentration of Atglistatin is very high in ATGLi liver, the authors should measure the lipolysis activity in liver extracts.

We previously demonstrated that addition of Atglistatin to liver extracts inhibits ATGL activity (Mayer et al., Nat Chem Biol, 2013, Figure attached). As suggested by the reviewer we performed TG hydrolase activity assays from liver extracts of Atglistatin fed animals but did not observe differences to activities found in livers of untreated animals (attached Fig. 1 and Supplemental Fig. 9b). This indicates that *in vivo* Atglistatin does not efficiently inhibit lipolysis. This inability of Atglistatin to inhibit hepatic ATGL and lipolysis all together may result from increased ATGL enzyme protein levels (Suppl. Fig. 9a), the transient nature of Atglistatin's action (Mayer et al., Nat Chem Biol, 2013), or the compensatory activity of alternative, currently unknown, TG hydrolases. The existence of alternative lipases is corroborated by the relatively small contribution of ATGL to the total lipolytic activity in liver lysates (see attached Fig.1). The various possibilities to explain our findings are discussed in the paper.

5) In the discussion, the authors want to discuss the paper published in Diabetes, 2009 by Ahmadian et al, which reported that overexpression of ATGL lead to improved insulin sensitivity.

We are sorry for this neglect and included the publication of Ahmadian et al. in the discussion of the manuscript.

Attached Figures

Attached Fig. 1: TG hydrolase activity of liver lysates from wild-type and ATGL deficient animals in the presence and in the absence of Atglistatin. Lysates of liver of wild-type (wt) and ATGL-ko mice were subjected for TG hydrolase activity assays in the presence or absence of 40 μ M Atglistatin using a substrate containing radiolabeled [9,10-3H(N)]-triolein. After incubation for 1h at 37°C, FA were extracted and quantitated by liquid scintillation counting. Data are presented as mean + S.D. (n=3 for each genotype; **, p < 0.01 for +/- Atglistatin and #, p < 0.01, for WT vs. ATGL-ko).

Attached Fig. 2: Leptin sensitivity and hypothalamic gene expression upon Atglistatin feeding. ATGLi and control mice were fasted for 6 h and re-fed after the injection of 5mg/kg leptin or Tris/Cl as control. At the indicated time-points food intake was assessed by subtracting food uneaten from the amount of food added to the bottom of cages. Hypothalamic mRNA expression was assessed of re-fed mice by RT-qPCR.

Reviewer #1 (Remarks to the Author)

Overall, the revised manuscript is technically sound, the conclusion are well supported, and limitations acknowledged.

The main strength of the report is the proof of principle that pharmacologic inhibition of ATGL can have beneficial effects on glucose homeostasis and reduce steatosis in liver and heart in mouse models.

The authors acknowledge that the concept of inhibiting adipocyte lipolysis to improve glucose homeostasis and reduce ectopic fat accumulation is not new. On the other hand, the concept that transient, tissue-specific lipase inhibition might be the key to the benefits of ATGL inhibition and explain differences in genetic and pharmacologic models is new and potentially important. Based on this work, it would seem that the benefits of ATGL inhibition in mice relate importantly to unique properties of the inhibitor, including PK, tissue distribution, route of administration, and effects on food intake. However, the mechanisms and sites of action underlying inactivation of ATGL presently remain a matter of informed speculation, and it is uncertain whether these properties can be reproduced in humans.

Reviewer #3 (Remarks to the Author)

Thanks for properly addressing my comments. The effects of Atglistatin on HFD are both exciting and intriguing.